# Association between *PAI-1* Polymorphisms and Ischemic Stroke in a South Korean Case-Control Cohort

**DOI:** 10.3390/ijms24098041

**Published:** 2023-04-28

**Authors:** Gun Ho Choi, Sung Hwan Cho, Hui Jeong An, Han Sung Park, Jeong Yong Lee, Eun Ju Ko, Seung Hun Oh, Ok Joon Kim, Nam Keun Kim

**Affiliations:** 1Department of Biomedical Science, College of Life Science, CHA University, Seongnam 13488, Republic of Korea; 2College of Medicine, Konyang University, 158 Gwanjeodong-ro, Seo-gu, Daejeon 35365, Republic of Korea; 3College of Life Science, Gangneung-Wonju National University, 7 Jukheon-gil, Gangneung 25457, Republic of Korea; 4Department of Neurology, CHA Bundang Medical Center, School of Medicine, CHA University, Seongnam 13496, Republic of Korea

**Keywords:** ischemic stroke, PAI-1, polymorphism, genetic association, Korean population

## Abstract

Stroke is the second leading cause of death in the world. Approximately 80% of strokes are ischemic in origin. Many risk factors have been linked to stroke, including an increased level of plasminogen activator inhibitor-1 (PAI-1). PAI-1 levels increase and remain elevated in blood during the acute phase of ischemic stroke, which can impair fibrinolytic activity, leading to coronary artery disease and arterial thrombotic disorders. Here, we present a case-control study of 574 stroke patients and 425 controls seen for routine health examination or treatment for nonspecific dizziness, nonorganic headache, or anxiety for positive family history of stroke at the Bundang Medical Center in South Korea. Polymorphisms in *PAI-1* were identified by polymerase chain reaction/restriction fragment length polymorphism analysis using genomic DNA. Specifically, three variations (−675 4G>5G, 10692T>C, and 12068G>A) were linked to a higher overall prevalence of stroke as well as a higher prevalence of certain stroke subtypes. Haplotype analyses also revealed combinations of these variations (−844G>A, −675 4G>5G, 43G>A, 9785A>G, 10692T>C, 11053T>G, and 12068G>A) that were significantly associated with a higher prevalence of ischemic stroke. To the best of our knowledge, this is the first strong evidence that polymorphic sites in *PAI-1* promoter and 3′-UTR regions are associated with higher ischemic stroke risk. Furthermore, the *PAI-1* genotypes and haplotypes identified here have potential as clinical biomarkers of ischemic stroke and could improve the prognosis and future management of stroke patients.

## 1. Introduction

In South Korea, stroke ranks as the second leading cause of death after cancer and surpasses heart disease in frequency; globally, it is the second leading cause of death [1,2,3,4]. The development of stroke is a complex, multifactorial, and polygenic disease influenced by various interactions between genes and the environment [5,6]. Approximately 80% of strokes are ischemic in origin [7]. Ischemic stroke has been linked to many risk factors, including alcohol abuse [8], coagulopathy (blood clotting disorder) [9], diabetes mellitus (DM) [10], fibromuscular dysplasia [11], family or personal history of stroke [12], high cholesterol [13], brain tumor [14], smoking [15], hypertension (HTN) [16], hyperlipidemia [17], and metabolic syndrome [18]. 

Ischemic stroke is caused by a sudden blockage of the blood supply in a brain artery [19]. This is most often caused by atherosclerosis or gradual cholesterol deposition. If the arteries become too narrow, blood cells collect and form blood clots, which can block arterial blood flow [20]. This can lead to brain damage, disability, and death and is one of the leading causes of morbidity and mortality worldwide [21]. Ischemic stroke can affect individuals of all ages, sex, and ethnicities, but certain risk factors, such as hypertension, diabetes, smoking, and obesity, can increase the likelihood of developing the condition [22,23].

Additionally, thrombophilia can increase susceptibility to ischemic stroke, and thrombotic responses are potential predisposing factors to ischemic stroke. Hence, it is worthwhile to investigate associations between genes related to thrombophilia, such as plasminogen activator inhibitor-1 (*PAI-1*), and ischemic stroke risk.

*PAI-1,* also known as SERPINE1, is located on the seventh human chromosome (7q21.3-22). *PAI-1* is the main inhibitor of tissue plasminogen activator and urokinase [24], which are responsible for converting plasminogen to the fibrinolysis enzyme plasmin [25]. Fibrinolysis is the process of fibrin degradation and is mainly regulated by PAI-1. Fibrinolysis retains blood vessel patency and degrades extracellular matrix [26].

Specific single-nucleotide polymorphisms (SNPs) are associated with the risk of stroke and other diseases. Promoters are among the many important genetic regions that regulate gene expression and contain functionally relevant polymorphic sites [27,28]. The genetic polymorphism rs1799889 (−675 4G>5G) is well known and in the *PAI-1* promoter. The rs1799889 ‘’4G’’ allele increases promoter activity and mRNA expression [29,30,31]. Another widely studied *PAI-1* polymorphism is rs2227631 (−844 G>A). The rs2227631 “A” allele is associated with increased transcriptional activity and an increased level of PAI-1 protein [32,33]. In addition, rs6092 (43G>A), rs2227 (694G>A), rs7242 (11053T>G), rs11178 (10692T>C), and rs1050955 (12068G>A) in the *PAI-1* 3′-UTR may affect the level of PAI-1 in the plasma [34]. In a recent study on PAI-1, significance thresholds were applied to the number of multitrait combinations. Of note, one of the newly discovered loci (venous thromboembolism) was associated with PAI-1 [35]. Bahrami et al. [36] conducted a systematic review and meta-analysis to determine if there is an association between PAI-1 rs1799889 polymorphism and arterial ischemic stroke. The meta-analysis showed a significant association between PAI-1 rs1799889 polymorphism and ischemic stroke in children, suggesting that this polymorphism may be a genetic risk factor for arterial ischemic stroke in this population. In another study, the genotype and allele frequencies of six risk genes were analyzed in both healthy individuals and patients with ischemic stroke. The results suggested that controlling clinical laboratory parameters, including common genotypes such as PAI-1, in patients with diabetes and hyperlipidemia may help preventative ischemic stroke [37].

Several polymorphisms within the PAI-1 gene have been found to influence PAI-1 levels, of which the most studied is the −675 4G/5G polymorphism of the promoter region (rs1799889). Previous studies indicated that the PAI-1 rs1799889 polymorphism alone does not increase the risk of ischemic stroke. However, the PAI-1 4G/4G genotype is significantly associated with increased triglyceride and decreased HDL cholesterol levels in the healthy group [38]. Several epidemiological studies have examined the relationship between PAI-1 rs1799889 polymorphism and the risk of ischemic stroke; yet, their findings have been inconsistent or even contradictory. For example, while Esparza-García et al., reported no association between PAI-1 rs1799889 polymorphism and an increased risk of ischemic stroke in Mexican populations [39], Xu et al., suggested that this polymorphism may be linked to a higher risk of ischemic stroke in Han Chinese [40]. Over the last few years, there have been investigations into the potential links between PAI-1 rs1799889 polymorphism and the risk of ischemic stroke. Nevertheless, due to the limited sample sizes, the findings from these studies have been inconclusive at times.

The 3’-UTR is an important region of mRNA transcripts that regulates gene expression through various mechanisms, such as polyadenylation, translation, and stability control, and by binding to regulatory proteins and miRNAs [41,42]. Several studies have suggested that polymorphisms in the 3’-UTR region of certain genes can have an impact on various diseases. For instance, a previous report indicated that rs7242 in the 3’-UTR of PAI-1 is linked with an increased risk of severe radiation pneumonitis in lung cancer patients [43]. Therefore, understanding the role of genetic variants in the 3’-UTR of PAI-1 is crucial for identifying the potential risk factors of various diseases. In this study, we investigated seven polymorphic sites in the promoter and 3’-UTR regions of PAI-1, including rs2227631, rs1799889, rs6092, rs2227694, rs11178, rs7242, and rs1050955, and we assessed their association with ischemic stroke risk in a Korean population.

## 2. Results

### 2.1. Baseline Characteristics of Ischemic Stroke Patients and Controls

The demographic and clinical characteristics of the ischemic stroke patients and controls in the study cohort are described in Table 1. Compared with controls, the patients with ischemic stroke had higher frequencies of metabolic syndrome (*p* < 0.0001), HTN (*p* < 0.0001), DM (*p* < 0.0001), hyperlipidemia (*p* < 0.047), high-density lipoprotein cholesterol (HDL-C) (*p* < 0.042), higher total homocysteine (*p* < 0.001) and fibrinogen (*p* < 0.016), lower activated partial thromboplastin time (aPTT) (*p* < 0.0004), and lower folate (*p* < 0.0001).

### 2.2. Genotype Frequencies of Seven PAI-1 Polymorphisms in Ischemic Stroke Patients and Controls

The *PAI-1* genotype frequencies observed in controls and patients with ischemic stroke were consistent with the Hardy–Weinberg equilibrium (*p* > 0.05). We calculated the adjusted odds ratios (AORs) from logistic regression analyses for age, sex, HTN, DM, smoking, and hyperlipidemia. The −675 [rs1799889] 4G>5G, 10692 [rs11178] T>C, and 12068 [rs1050955] G>A genotypes of *PAI-1* were significantly associated with ischemic stroke risk, whereas the −844 [rs2227631] G>A, 43 [rs6092] G>A, 9785 [rs2227694] A>G, and 11053 [rs7242] T>G polymorphisms were not (Table 2). However, after multiple comparisons, only subjects with *PAI-1* rs1050955 G>A, AA had a 1.784-fold increased risk of ischemic stroke relative to patients with the GG genotype (Table 2). In addition, the present case-control study analyzed the association between these seven *PAI-1* SNPs and ischemic stroke risk in the South Korean patient cohort described in Figure 1.

### 2.3. Genotype Frequencies of Seven PAI-1 Polymorphisms Stratified by Ischemic Stroke Subtype

We classified the patients with stroke into three subgroups depending on the stroke etiology: large-artery disease (LAD), small-vessel disease (SVD), or cardioembolism (CE). The LAD subtype was significantly associated with the 12,068 (rs1050955) polymorphism. However, after multiple comparisons, only the *PAI-1* rs1050955 dominant genotype was associated with a 1.792-fold increased risk of LAD-subtype stroke relative to the GG genotype, and the statistical significance was borderline (*p* < 0.05) (Table 3). The SVD subtype was significantly associated with the 10692(rs11178) polymorphism. However, after multiple comparisons, only the *PAI-1* rs11178 CC genotype was associated with a 2.238-fold increased risk of SVD-subtype stroke relative to the TT genotype, and the statistical significance was borderline (*p* < 0.05) (Table 3). CE stroke subtype was significantly associated with rs1799889, although after multiple comparisons, only subjects with the *PAI-I* rs1799889 recessive genotype had a 2.529-fold risk relative to patients with 4G4G genotype; statistical significance was borderline (*p* < 0.05) (Table 3).

### 2.4. Analysis of PAI-1 Haplotypes in Ischemic Stroke and Metabolic Syndrome Patients and Controls

We analyzed the association between *PAI-1* haplotypes and ischemic stroke and metabolic syndrome in the study cohort using multifactor dimensionality reduction (MDR), and the results are presented in Table 4. The following haplotypes demonstrated a significant association with an increased risk of stroke in the study cohort: A/5G/G/G/C/T/A (rs2227631/rs1799889/rs6092/rs2227694/rs11178/rs7242/rs1050955), A/5G/G/C/T/A (rs2227631/rs1799889/rs6092/rs11178/rs7242/rs1050955), A/5G/C/T/A, G/4G/T/G/G, A/5G/T/G/G (rs2227631/rs1799889/rs11178/rs7242/rs1050955), A/5G/G/C (rs2227631/rs1799889/rs6092/rs11178), A/5G/C, A/5G/T (rs2227631/rs1799889/rs11178), and A/5G (rs2227631/rs1799889).

The PAI-1 haplotypes associated with decreased risk of stroke included G/4G/A/T and A/4G/G/T (rs2227631/rs1799889/rs6092/rs11178). The haplotypes associated with decreased risk of ischemic stroke and metabolic syndrome were A/4G/A/T (rs2227631/rs1799889/rs6092/rs11178) (Table 4). An analysis of linkage disequilibrium (LD) of loci −844 (rs2227631)/−675 (rs1799889)/43 (rs6092)/9785 (rs2227694)/10,692 (rs11178)/11,053 (rs7242)/12,068 (rs1050955) in ischemic stroke patients and controls is shown in Appendix A. In stroke patients, LD was detected between loci −844 and −675 (D’ = 0.759, logarithmic odds (LOD) = 61.25, r2 = 0.230), 9785 and 11,053 (D’ = 0.764, LOD = 6.020, r2 = 0.021), 9785 and 12,068 (D’ = 0.761, LOD = 3.670, r2 = 0.015), 10,692 and 11,053 (D’ = 0.916, LOD = 168.0, r2 = 0.545), and 11,053 and 12,068 (D’ = 0.831, LOD = 121.74, r2 = 0.430). All haplotype data are presented in Appendix A.

### 2.5. Combined Genotype Analysis for PAI-1 in Metabolic Syndrome and Stroke Patients and Controls

Several PAI-1 alleles were associated with an increased risk of ischemic stroke. These included ‘GA/5G5G’ (rs2227631/rs1799889), ‘AA/TT’ (rs2227631/rs7242), ‘GA/CC’ (rs2227631/rs11178), ‘AA/TC’ (rs2227631/rs11178), ‘5G5G/TC’ (rs1799889/rs11178), ‘5G5G/CC’ (rs1799889/rs11178), ‘5G5G/AA’ (rs1799889/rs1050955), ‘GA/AA’ (rs6092/rs1050955), ‘GG/GA’ (rs2227694/rs1050955), ‘GG/AA’ (rs2227694/rs1050955), ‘TC/GA’ (rs11178/rs1050955), and ‘CC/AA’ (rs11178/rs1050955) (Table 5). In contrast, the following PAI-1 alleles were associated with decreased stroke risk: ‘AA/TT’ (rs2227631/rs11178), ‘4G4G/GA’ (rs1799889/rs6092), ‘GA/TT’ (rs6092/rs11178), ‘GA/GG’ (rs6092/rs7242), and ‘GA/GG’ (rs6092/rs1050955). According to unadjusted logistic regression analysis, an increased risk of metabolic syndrome was associated with PAI-1 alleles ‘GA/5G5G’ (rs2227631/rs1799889), ‘GA/GG’ (rs2227631/rs6092), ‘AA/GA’ (rs2227631/rs2227694) and ‘GA/AA’ (rs2227631/rs1050955). Conversely, decreased risk of metabolic syndrome was associated with PAI-1 alleles ‘4G4G/GA’ (rs1799889/rs6092), ‘GG/TC’ (rs6092/rs11178), ‘TC/TG’ (rs11178/rs7242), and ‘CC/TT’ (rs11178/rs7242) (Table 5). Supporting data relevant to Table 5 are shown in Appendix A. In addition, we conducted ANOVA analyses of potential risk factors for stroke. The results showed that the rs1799889 recessive model was associated with vitamin B12 levels (*p* = 0.023; Appendix A).

### 2.6. Ischemic Stroke Incidence Stratified by Sex, Advanced Age, Hypertension, Diabetes Mellitus, Hyperlipidemia, Smoking, Folate, and Homocysteine

*PAI-1* −675 4G5G + 5G5G with HTN was two-fold more likely to increase ischemic stroke than the 4G4G genotype without HTN and was three-fold more likely to increase ischemic stroke risk in patients with a folate level ≤ 3.57 nmol/L (Figure 2). In addition, the *PAI-1* 12,068 GA + AA genotype with DM was two-fold more likely to increase the risk of ischemic stroke than the GG genotype without DM (Figure 2).

### 2.7. Baseline Characteristics, Genotype Frequencies and Subtypes between Ischemic Stroke Patients and Control Subjects in Subgroups 1 and 2

A total of 999 individuals (574 patients with stroke and 425 control subjects) from two different case-control samples, sample 1 (recruitment period: 2001 to 2006) and sample 2 (recruitment period: 2007 to 2010), were analyzed according to recruitment duration (Appendix A). The baseline characteristics between ischemic stroke patients and control subjects in samples 1 and 2 in the study cohort are described in Appendix A. Compared with controls, the patients with ischemic stroke had higher frequencies of metabolic syndrome (*p* < 0.0001), HTN (*p* < 0.0001), DM (*p* < 0.001, sample 1; *p* < 0.0001, sample 2), hyperlipidemia (*p* < 0.004, sample 1), folate (*p* < 0.0001 sample 1; *p* < 0.0006, sample 2), and total homocysteine (*p* < 0.001, sample 1) (Appendix A). The genotype frequency of the PAI-1 12068G>A polymorphisms were significantly different between the control and ischemic stroke groups in sample 2 (Appendix A). The LAD of the PAI-1 −844G>A polymorphisms were significantly different between the control and ischemic stroke groups in sample 2 (Appendix A).

### 2.8. Clinical Variables in Ischemic Stroke Patients Stratified by PAI-1 −844GA/−675 5G Genotype

Patients with the *PAI-1* ‘GA/5G5G’ (rs2227631/rs1799889) genotype had higher total cholesterol, BMI, and uric acid than patients with the *PAI-1* ‘GG+4G4G’ (rs2227631/rs1799889) genotype (*p* = 0.013, *p* = 0.011, and *p* = 0.041, respectively; Appendix A).

## 3. Discussion

In this case-control study, seven polymorphic sites in *PAI-1* were identified, and their associations with increased or decreased risk of ischemic stroke were investigated in a South Korean cohort with 574 stroke patients and 425 controls. The results provide evidence for strong associations between rs1799889, rs11178, and rs1050955 polymorphisms in *PAI-1* and risk of ischemic stroke. Furthermore, the rs1799889 ‘5G5G’, rs11178 ‘CC’, and rs1050955 ‘AA’ genotypes were associated with the CE, SVD, and LAD ischemic stroke subtypes, respectively. In addition, the rs1799889, rs11178, and rs1050955 polymorphisms had synergic effects on ischemic stroke risk when combined. Haplotype analyses showed that the rs2227631 ‘A’, rs1799889 ‘5G’, rs11178 ‘C’, and rs1050955 ‘A’ alleles were associated with increased risk of ischemic stroke. Furthermore, the rs2227631/rs1799889 ‘GA/5G5G’ combination genotype was associated with increased ischemic stroke risk. In contrast, the rs1799889 ‘4G’ allele was associated with decreased ischemic stroke risk in most haplotypes.

Previous studies of *PAI-1* have identified associations with vascular diseases [44,45]. The common rs1799889 ‘4G’ allele in the promoter region of *PAI-1* is linked to increased transcription of *PAI-1* and is considered an independent marker of the inhibition of plasminogen activation [46]. In contrast to a previous study that linked rs1799889 ‘4G’ to increased risk of ischemic stroke [44], our data as well as previously presented data [45] suggest that rs1799889 ‘4G’ is associated with a reduced risk of ischemic stroke. The mechanism by which rs1799889 ‘4G’ increases the expression of *PAI-1* was inferred from the fact that the rs1799889 ‘5G’ allele binds upstream stimulatory factor (USF-1) with relatively lower efficiency than it binds the rs1799889 ‘4G’ allele [47]. Half the Korean population is heterozygous (‘4G/5G’) at rs1799889, and 25% of the population carries the homozygous ‘5G/5G’ genotype at rs1799889. Functional studies demonstrated differential binding of USF-1 at this site [48]. We hypothesize that the effect of the rs1799889 polymorphism on stroke risk is also modulated by other polymorphisms or genes.

The rs2227631 polymorphism is another common SNP in the *PAI-1* promoter region. The rs2227631 ‘A’ allele is associated with increased promoter activity [34]. *PAI-1* promoter-region polymorphisms including rs2227631, rs1799889 single genotypes, and haplotypes have been widely studied. Previous studies have investigated the associations between these SNPs in *PAI-1* and risk of inflammatory diseases, vascular diseases, myocardial infarction, and metabolic syndrome [49,50,51,52,53,54]. In the present study, we did not find an association between rs2227631 and ischemic stroke; however, we found that the rs2227631/rs1799889 ‘GA/5G5G’ genotype and ‘A/5G’ haplotype were strongly associated with risk of ischemic stroke and metabolic syndrome. Moreover, the rs2227631/rs1799889 ‘GA/5G5G’ genotype was associated with higher total cholesterol, uric acid, and BMI. High BMI is a direct risk factor for metabolic syndrome. High total cholesterol is an indirect risk factor for metabolic syndrome and a direct risk factor for ischemic stroke [55,56,57,58]. Therefore, we hypothesize that the combination rs2227631/rs1799889 ‘AA/5G5G’ genotype in the *PAI-1* promoter region contributes to and should be considered when managing the risk of ischemic stroke. A recent study also reported that genetic polymorphisms in *PAI-1* were associated with disease states and folate and homocysteine levels. More specifically, *PAI-1* −675 5G5G (AOR, 3.302; *p* = 0.017) and 43GA + AA (AOR, 2.087; *p* = 0.042) genotype frequencies showed significant associations with increased prevalence of osteoporotic vertebral compression fractures (OVCFs) in postmenopausal women [59].

*PAI-1* polymorphisms also modulate the risk of colorectal cancer (CRC) in a manner that is dependent on other clinical pathologic factors such as age, presence or absence of HTN, presence or absence of DM, obesity, folate levels, and homocysteine levels. The *PAI-1*−675 4G5G + 5G5G genotype was associated with a higher risk of CRC than the 4G4G genotype without HTN or DM; however, the *PAI-1* +11053TT + TG genotype with HTN or without DM had a lower risk of CRC than the GG genotype. Moreover, at folate levels ≥3.8 ng/mL, the *PAI-1* −675 4G5G + 5G5G genotype positively correlated with CRC risk, while the *PAI-1* 11053GG genotype inversely correlated with CRC risk. These findings suggest that genetic factors and comorbidities/pathologies including metabolic diseases interact with each other while modulating CRC risk. These data may implicate *PAI-1* polymorphisms in linking and/or modulating the relationship between metabolic syndrome and cancer [60].

Another study identified six risk factors for metabolic syndrome including HTN, DM, hyperlipidemia, BMI ≥ 25 kg/m^2^, TG ≥ 150 mg/dL, and HDL < 40 mg/dL (male) or <50 mg/dL (female), which, combined with the *PAI-1* −675 4G5G + 5G5G genotype, conferred a significantly increased AOR of metabolic syndrome (AOR = 2.780, 3.266, 1.779, 4.050, 1.714, and 6.781, respectively) relative to the PAI-1−675 4G4G genotype without these six predisposing risk factors [61].

These data suggest that the risk of coronary artery disease, CRC, and OVCF, which can reflect and be linked to underlying metabolic diseases (e.g., DM, HTN, and folate), may be further modulated by the *PAI-1*, genotype leading to a much higher risk of disease. Our data suggest a similar set of relationships between DM, HTN, folate levels, risk of ischemic stroke, and the *PAI-1* genotype. In particular, a reduced risk of metabolic syndrome and stroke is associated with *PAI-1* SNPs in the promoter region, −844G>A, −6754G>5G, exon region (43G>A), and the 3′-UTR region SNP combination A-4G-A-T (MetS; OR: 0.073, FDR-p: 0.006) (stroke; OR: 0.141, FDR-p: 0.001).

Post-transcriptional regulation of gene expression frequently involves sequence-specific interactions between miRNAs and the 3′-UTR region of the target mRNA transcript [56,57,62]. Our study confirms the association between promoter and 3’-UTR region polymorphisms and stroke disease and provides a new understanding of the genetic basis [63]. Promoter polymorphisms can affect gene transcription and expression levels, while 3’-UTR region polymorphisms can affect mRNA stability and translation efficiency [64]. Identifying these functional SNPs and their association with stroke risk can aid in the development of personalized prevention and treatment strategies for individuals at risk of developing stroke [65]. Additionally, studying these SNPs can provide insights into the mechanisms underlying stroke development and may lead to the development of novel therapies [66].

Furthermore, a genetic polymorphism in the 3′-UTR region can alter the affinity of a miRNA for its target binding sequence. Previous studies demonstrated that *PAI-1* SNPs rs1799889 and rs2227631 in the promoter region and rs11178 in the 3′-UTR were associated with risk for osteonecrosis of the femoral head [67]. Another report predicted that combinations of *PAI-1* rs2227631, rs1799889, and rs7242 would have synergistic effects on the risk of recurrent pregnancy loss [68]. Most analyses of *PAI-1* polymorphism as a risk factor for stroke have focused on rs1799889 and rs2227631 in the *PAI-1* promoter region [29,30,31,32,33,34]. A meta-analysis of 44 case-control studies, representing a total of 8620 cases and 10,260 controls, showed a significant association between *PAI-1* rs1799889 and the risk of ischemic stroke. This meta-analysis suggested that *PAI-1* rs1799889 is associated with increased risk of ischemic stroke, especially in Asian and mixed populations [69].

Another meta-analysis demonstrated a significant association between risk of atherosclerosis and *PAI-1* rs2227631 in the dominant model as well as significant associations between *PAI-1* rs1799889 and the dominant, recessive, and allele models. Moreover, the rs1799889 polymorphism was significantly correlated with the risk of atherosclerosis in Asians (dominant model: 95% CI 1.10–1.83; allele model: 95% CI 1.03–1.41) and Caucasians (recessive model: 95% CI 0.87–0.97; allele model: 95% CI 1.01–1.12). These findings indicate that *PAI-1* rs2227631 and rs1799889 have potential as genetic biomarkers of atherosclerosis [70].

A recent meta-analysis provided evidence for a significant association between *PAI-1* 4G/5G and an increased risk of adult but not pediatric ischemic stroke (adult: 4G/4G vs. 4G/5G + 5G/5G, OR =1.21, 95% CI = 1.04–1.42). In the subgroup analysis, a significant association was detected among Asians (4G/4G vs. 4G/5G + 5G/5G, OR = 1.45, 95% CI = 1.14–1.85) [71]. Our data provide strong evidence that polymorphisms in the promoter and polymorphism combinations in the 3′-UTR of *PAI-1* are associated with ischemic stroke risk.

We could not demonstrate that *PAI-1* 3′-UTR polymorphisms alter the binding affinity of miRNA to the 3′-UTR; however, recent data showed that miRNA binding affinity is only sensitive to one SNP in its target binding sequence, which could subsequently influence the expression of PAI-1 protein [72]. Specifically, miR-421 and miR-30c bind with higher affinity to the rs1050955 ‘A’ allele, resulting in a decreased expression of *PAI-1* [73].

Although we identified positive genetic associations between *PAI-1* polymorphisms and the risk of ischemic stroke, there are some limitations to our study. First, the mechanisms by which the *PAI-1* polymorphisms modulate the risk of ischemic stroke are still unclear. Second, the environmental risk factors that influence the risk of stroke need to be further investigated. Third, the study population exclusively resided in South Korea and may lack diversity. Future studies should include a more heterogeneous cohort to determine whether similar results are obtained in diverse patient subgroups. If functional research definitively identifies a causal role for *PAI-1* in the pathogenesis of ischemic stroke, it may be possible to prevent or reduce risk of stroke by altering PAI-1 expression or activity. The effects of polymorphisms and transcription factors that regulate the *PAI-1* promoter and 3’-UTR also warrant further study.

There are several additionally limitations to our study. Functional studies for PAI-1 SNP were not performed to elucidate the stroke-related pathogenesis including PAI-1 activity. Although several studies have reported an association between PAI-1 polymorphisms and stroke, few have evaluated the pathogenesis by which PAI-1 polymorphisms affect stroke in Korean patients. This study cannot, therefore, propose a detailed pathogenesis by which PAI-1 polymorphism affects stroke. This study was a hospital-based case-control study, which had a relatively small sample size of individual stroke subtypes. Although a population-based study may be optimal to reduce the selection bias, it is difficult to obtain sufficient numbers of stroke incidents among the cohort because the estimated annual incidence of stroke is known to be low in the general population. Our results cannot be generalized to other races because racial variability in stroke subtype and genotype frequencies may produce different results.

In summary, we identified associations between rs2227631, rs1799889, rs6092, rs2227694, rs11178, rs7242, and rs1050955 polymorphisms in *PAI-1* and the risk of ischemic stroke in a South Korean study cohort. A recent meta-analysis indicated that studies on the association between *PAI-1* polymorphisms and ischemic stroke have typically focused on the −675 4G/5G and −844 G/A SNPs in the *PAI-1* promoter region. However, our study indicates that SNPs in the 3’-UTR region, which plays a functional role and regulates *PAI-1* expression, also modulate risk of stroke. Specifically, we report the ischemic-stroke-induced association of *PAI-1* rs11178 (10692) T>C, rs1050955 (12068) G>A. Thus, polymorphisms in the *PAI-1* promoter and 3’-UTR both contribute to ischemic stroke risk and might be useful as biomarkers of the risk of ischemic stroke. Our findings can be used to plan for patients at risk of developing ischemic stroke by performing genetic testing to confirm the presence of the polymorphism. Additionally, further research may be conducted to develop targeted therapies to reduce the risk of ischemic stroke in patients with PAI-1 genetic polymorphism. Additional epidemiological studies on more heterogeneous study populations are needed, as these studies will improve our understanding of the relationships between *PAI-1* polymorphisms and the relative risk of ischemic stroke.

## 4. Materials and Methods

### 4.1. Study Population

We recruited study participants from 2000 to 2008 from the Seoul and Kyeonggi-do provinces of South Korea. The Institutional Review Board of CHA Bundang Medical Center approved our genetic study in June 2000. Five hundred seventy-four consecutive patients with ischemic stroke referred from the Department of Neurology at CHA Bundang Medical Center, CHA University, were enrolled in the study. Ischemic stroke was defined as a stroke (characterized by rapidly developing clinical symptoms and signs of focal or global loss of brain function) with evidence of cerebral infarction in clinically relevant areas of the brain according to brain magnetic resonance imaging (MRI) and electrocardiography. Plasma samples were obtained within 48 h of hospital admission from stroke patients or age- and sex-matched healthy donors. Based on clinical manifestations and neuroimaging data, two neurologists classified all ischemic strokes into four etiologic subtypes using the criteria from the Trial of Org 10,172 in Acute Stroke Treatment (TOAST) [74] as follows: 1) LAD is characterized by an infarction lesion ≥15 mm in diameter shown by MRI and significant (>50%) stenosis of a major brain artery or branch cortical artery detected by cerebral angiography along with symptoms associated with that arterial territory; 2) SVD is characterized by an infarction lesion <15 mm and ≥5 mm in diameter shown by MRI and classic lacunar syndrome, without evidence of cerebral cortical dysfunction or detectable cardiac sources of embolism; 3) CE, or arterial occlusions, presumably caused by an embolus arising in the heart, are detected by cardiac evaluation; and 4) undetermined etiology or multiple etiologies. The frequencies of etiological subtypes in the study sample were 51% (n = 200) LAD, 36% (n = 140) SVD, and 13% (n = 52) CE. These proportions are similar to previously reported values for the Korean population. Strokes due to single and multiple (≥2 lesions) SVD were distinguished by brain MRI scan. The size and site of cerebral infarction were documented only by MRI.

We enrolled 425 control subjects during the first 5 years of the enrollment period, who underwent health examinations such as biochemistry testing, an electrocardiogram, or brain MRI. The controls did not have a recent history of cerebrovascular disease or myocardial infarction and were matched by sex and age to the patients with stroke. Exclusion criteria were the same as those used in the patient group. Hypertension was defined by current use of hypertensive medications or systolic pressure >140 mmHg and diastolic pressure >90 mmHg on >1 occasion. Diabetes was defined by current use of diabetic medications or fasting plasma glucose >126 mg/dL (7.0 mmol/L). Smoking referred to current smoking only. Hyperlipidemia was defined by a history of treatment with antihyperlipidemic agents or a fasting serum total cholesterol level ≥240 mg/dL.

Patients were diagnosed with metabolic syndrome if they possessed three or more of the following clinical factors: body mass index (BMI) ≥25.0 kg/m^2^; triglycerides ≥150 mg/dL; high-density lipoprotein cholesterol (HDL-C) ≤40 mg/dL in men or ≤50 mg/dL in women; blood pressure ≥140/90 mmHg or current use of hypertensive medication; and fasting plasma glucose (FPG) ≥110 mg/dL or current use of insulin or hypoglycemic medication. Hyperlipidemia was defined as fasting total serum cholesterol level ≥240 mg/dL or a prior history of treatment with lipid-lowering medication. Data on current smoking status were also collected.

### 4.2. Determination of PAI-1 Genotypes

Genomic DNA was extracted from blood leukocytes using a G-DEX blood extraction kit (Intron Inc., Seongnam, Korea). Nucleotide changes were determined by polymerase chain reaction (PCR) restriction fragment length polymorphism (RFLP) analysis using the isolated genomic DNA as a template. PCR was performed using the primers listed in Appendix A.

### 4.3. Statistical Analyses

The associations between ischemic stroke and PAI-1 genotypes were estimated by computing ORs and 95% CIs from Fisher exact tests. AORs for *PAI-1* polymorphisms were determined from multiple logistic regression analyses using sex, age, DM, HTN, hyperlipidemia, and smoking as covariates. Analyses were performed using GraphPad Prism 4.0 (GraphPad Software Inc., San Diego, CA, USA) and Medcalc version 12.7.1.0 (Medcalc Software, Mariakerke, Belgium). Stratification analysis was used to classify the strokes into subgroups according to the size of the occluded vessel. StatsDirect Statistical Software (Version 2.4.4; StatsDirect Ltd., Altrincham, UK) was used to calculate the ORs and 95% CIs in the stratification analysis. Haplotypes for multiple loci were estimated using the expectation-maximization algorithm with SNPAlyze (Version 5.1; DYNACOM Co, Ltd., Yokohama, Japan). Genetic interaction analysis was performed with the open-source MDR software package (v.2.0), available from www.epistasis.org (accessed on 12 May 2022). Survival curves were created through Cox proportional hazards regression, and log-rank tests were used to assess the significance of differences between groups. Cox regression models were used to analyze the independent prognostic importance of various markers, with results adjusted for age, sex, DM, HTN, hyperlipidemia, and smoking. Hazard ratios (HRs) are presented with 95% CIs using Medcalc version 12.7.1.0.

## Figures and Tables

**Figure 1 ijms-24-08041-f001:**
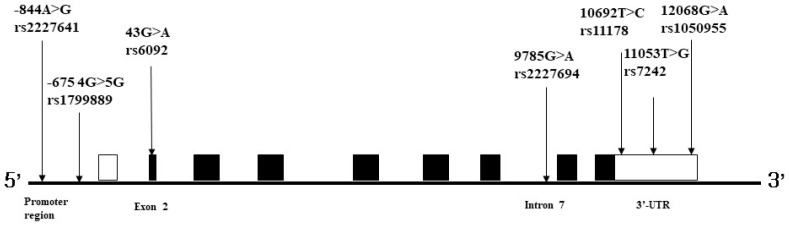
Structural characteristics of *PAI-1* and *PAI-1* polymorphic sites. Notes: *PAI-1* consists of nine exons. Seven SNPs in the promoter and 3′-UTR are indicated and are as follows: −844 [rs2227631] G>A, −675 [rs1799889] 4G>5G, 43 [rs6092] G>A, 9785[rs2227694] A>G, 10,692 [rs11178] T>C, 11,053 [rs7242] T>G and 12,068 [rs1050955] G>A. PAI-1 −675 4G>5G, 10692T>C, and 12068G>A.

**Figure 2 ijms-24-08041-f002:**
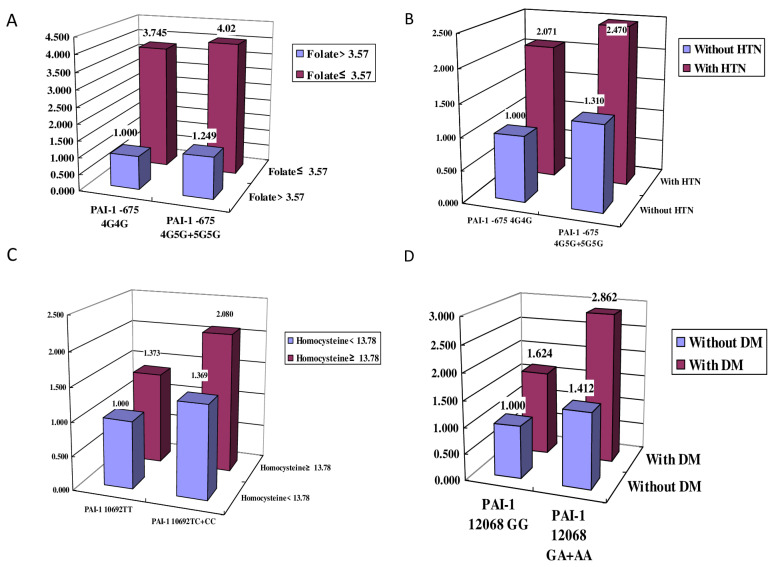
Ischemic stroke incidence (odds ratio) and interactions between genes and environmental factors such as (**A**) folate (FA), (**B**) hypertension (HTN), (**C**) homocysteine, and (**D**) diabetes mellitus (DM).

**Table 1 ijms-24-08041-t001:** Baseline characteristics of ischemic stroke patients and controls.

Characteristic	Controls (n = 425)	Stroke Patients (n = 574)	*p* ^a^
Male (%)	173 (40.7)	234 (40.8)	1.000
Age (years, mean ± SD)	62.62 ± 10.77	62.63 ± 11.56	0.986
Smoking (%)	138 (32.5)	206 (35.9)	0.448
Metabolic syndrome (%)	48 (11.3)	187 (32.6)	<0.0001
Hypertension (%)	170 (40.0)	362 (63.1)	<0.0001
Diabetes mellitus (%)	56 (13.2)	146 (25.4)	<0.0001
Hyperlipidemia (%)	101 (23.8)	181 (31.5)	0.047
BMI (kg/m^2^, mean ± SD)	24.23 ± 3.24	24.15 ± 3.09	0.723
HDL-C (mg/dl, mean ± SD)	46.40 ± 12.71	44.81 ± 14.26	0.042
Homocysteine (μmol/L, mean ± SD)	9.96 ± 4.10	11.14 ± 6.73	0.001
Folate (nmol/L, mean ± SD)	8.88 ± 7.88	7.01 ± 5.19	<0.0001
Vitamin B12 (pg/mL, mean ± SD)	742.81 ± 666.61	739.84 ± 618.84	0.943
Total cholesterol (mg/dL, mean ± SD)	194.03 ± 37.81	191.41 ± 40.25	0.301
Triglyceride (mg/dL, mean ±SD)	145.97 ± 88.82	151.56 ± 101.30	0.369
PLT (10^3^/μL, mean ± SD)	243.00 ± 66.59	249.55 ± 87.81	0.202
PT (s, mean ± SD)	11.76 ± 0.81	11.79 ± 0.98	0.720
aPTT (s, mean ± SD)	33.36 ± 18.28	30.44 ± 4.43	0.0004
Fibrinogen (mg/dL, mean ± SD)	393.29 ± 119.56	422.22 ± 129.11	0.016
Antithrombin (%, mean ± SD)	94.31 ± 43.40	94.28 ± 18.69	0.990
BUN (mg/dL, mean ± SD)	15.85 ± 4.98	16.00 ± 7.35	0.706
Uric acid (mg/dL, mean ± SD)	4.63 ± 1.44	4.66 ± 1.57	0.735

SD, standard deviation; BMI, body mass index; HDL-C, high-density lipoprotein cholesterol; PLT, platelet; PT, prothrombin time; aPTT, activated partial thromboplastin time; BUN, blood urea nitrogen. ^a^
*p*-values were calculated by two-sided Student’s *t*-test for continuous variables and chi-square test for categorical variables.

**Table 2 ijms-24-08041-t002:** Genotype frequencies of seven *PAI-1* polymorphisms in ischemic stroke patients and controls.

Characteristics	Controls (n = 425)	Stroke Patients (n = 574)	COR (95% CI)	*p*	*FDR-P*	AOR (95% CI) ^a^	*p*	*FDR-P*
*PAI-1* −844[rs2227631]G>A								
GG	162 (38.1)	201 (35.0)	1.000 (reference)			1.000 (reference)		
GA	191 (44.9)	292 (50.9)	1.232 (0.935–1.624)	0.138	0.327	1.122 (0.840–1.499)	0.436	0.610
AA	72 (16.9)	81 (14.1)	0.907 (0.621–1.325)	0.613	0.715	0.940 (0.630–1.402)	0.762	0.762
Dominant			1.143 (0.881–1.483)	0.314	0.366	1.077 (0.820–1.414)	0.595	0.694
Recessive			0.806 (0.570–1.138)	0.220	0.384	0.867 (0.604–1.244)	0.438	0.526
HWE *P*	0.224	0.126						
*PAI-1* −675[rs1799889]4G>5G								
4G4G	182 (42.8)	214 (37.3)	1.000 (reference)			1.000 (reference)		
4G5G	188 (44.2)	258 (44.9)	1.119 (0.852–1.470)	0.419	0.489	1.093 (0.821–1.455)	0.543	0.634
5G5G	55 (12.9)	102 (17.8)	1.579 (1.078–2.312)	0.019	0.067	1.646 (1.095–2.473)	0.017	0.051
Dominant			1.223 (0.947–1.579)	0.123	0.179	1.197 (0.916–1.564)	0.189	0.331
Recessive			1.489 (1.045–2.121)	0.028	0.098	1.513 (1.045–2.189)	0.028	0.087
HWE *P*	0.378	0.117						
*PAI-1* 43[rs6092]G>A								
GG	348 (81.9)	491 (85.5)	1.000 (reference)			1.000 (reference)		
GA	70 (16.5)	78 (13.6)	0.790 (0.556–1.122)	0.187	0.327	0.831 (0.575–1.202)	0.326	0.571
AA	7 (1.6)	5 (0.9)	0.506 (0.159–1.608)	0.248	0.434	0.552 (0.166–1.835)	0.332	0.498
Dominant			0.764 (0.544–1.073)	0.120	0.179	0.809 (0.566–1.155)	0.243	0.340
Recessive			0.525 (0.165–1.665)	0.274	0.384	0.572 (0.172–1.906)	0.363	0.526
HWE *P*	0.121	0.337						
*PAI-1* 9785[rs2227694] A>G								
GG	396 (93.2)	531 (92.5)	1.000 (reference)			1.000 (reference)		
GA	28 (6.6)	42 (7.3)	1.119 (0.682–1.836)	0.658	0.658	1.064 (0.634–1.783)	0.815	0.815
AA	1 (0.2)	1 (0.2)	0.746 (0.047–11.960)	0.836	0.836	N/A	0.993	N/A
Dominant			1.106 (0.678–1.803)	0.687	0.687	1.022 (0.613–1.704)	0.934	0.934
Recessive			0.740 (0.046–11.865)	0.832	0.832	N/A	0.993	N/A
HWE *P*	0.503	0.859						
*PAI-1* 10692[rs11178]T>C								
TT	159 (37.4)	185 (32.2)	1.000 (reference)			1.000 (reference)		
TC	204 (48.0)	287 (50.0)	1.209 (0.916–1.596)	0.180	0.327	1.311 (0.979–1.755)	0.069	0.242
CC	62 (14.6)	102 (17.8)	1.414 (0.967–2.068)	0.074	0.173	1.504 (1.003–2.255)	0.048	0.096
Dominant			1.257 (0.966–1.635)	0.089	0.179	1.348 (1.022–1.777)	0.034	0.119
Recessive			1.265 (0.897–1.785)	0.180	0.384	1.264 (0.882–1.812)	0.202	0.404
HWE *P*	0.793	0.609						
*PAI-1* 11053[rs7242]T>G								
TT	108 (25.4)	166 (28.9)	1.000 (reference)			1.000 (reference)		
TG	217 (51.1)	267 (46.5)	0.754 (0.558–1.019)	0.066	0.327	0.783 (0.573–1.071)	0.126	0.294
GG	100 (23.5)	141 (24.6)	0.910 (0.640–1.292)	0.596	0.715	0.885 (0.611–1.281)	0.517	0.620
Dominant			0.803 (0.605–1.065)	0.128	0.179	0.820 (0.610–1.101)	0.187	0.331
Recessive			1.088 (0.812–1.459)	0.571	0.666	1.048 (0.771–1.423)	0.766	0.766
HWE *P*	0.657	0.104						
*PAI-1* 12068[rs1050955] G>A								
GG	163 (38.4)	190 (33.1)	1.000 (reference)			1.000 (reference)		
GA	210 (49.4)	283 (49.3)	1.156 (0.878–1.522)	0.301	0.421	1.330 (0.993–1.783)	0.056	0.242
AA	52 (12.2)	101 (17.6)	1.666 (1.123–2.472)	0.011	0.067	1.784 (1.177–2.704)	0.006	0.036
Dominant			1.257 (0.968–1.633)	0.086	0.179	1.421 (1.078–1.875)	0.013	0.091
Recessive			1.532 (1.068–2.198)	0.021	0.098	1.520 (1.043–2.216)	0.029	0.087
HWE *P*	0.212	0.804						

COR, crude odds ratio; AOR, adjusted odds ratio; HWE, Hardy–Weinberg equilibrium; 95% CI, 95% confidence interval; N/A, not applicable; FDR, false discovery rate. ^a^ Adjusted for age, sex, hypertension, diabetes mellitus, hyperlipidemia, and smoking.

**Table 3 ijms-24-08041-t003:** Genotype frequencies of seven *PAI-1* polymorphisms stratified by ischemic stroke subtype.

Characteristics	Controls (n = 425)	LAD (n = 200)	AOR (95% CI)^a^	*p*	*FDR-P*	SVD (n = 140)	AOR (95% CI) ^a^	*p*	*FDR-P*	CE (n = 52)	AOR (95% CI) ^a^	*p*	*FDR-P*
*PAI-1* −675 [rs1799889]4G>5G													
4G4G	182 (42.8)	74 (37.0)	1.000 (reference)			55 (39.3)	1.000 (reference)			17 (32.7)	1.000 (reference)		
4G5G	188 (44.2)	92 (46.0)	1.167 (0.790–1.725)	0.437	0.653	59 (42.1)	0.937 (0.602–1.458)	0.772	0.901	21 (40.4)	1.208 (0.610–2.391)	0.588	0.735
5G5G	55 (12.9)	34 (17.0)	1.652 (0.937–2.914)	0.083	0.212	26 (18.6)	1.697 (0.930–3.098)	0.085	0.142	14 (26.9)	2.790 (1.239–6.279)	0.013	0.065
Dominant			1.240 (0.859–1.790)	0.251	0.407		1.069 (0.709–1.610)	0.751	0.860		1.566 (0.843–2.909)	0.155	0.813
Recessive			1.426 (0.870–2.338)	0.160	0.458		1.628 (0.951–2.788)	0.076	0.190		2.529 (1.254–5.099)	0.010	0.050
*PAI-1* 10692 [rs11178]T>C													
TT	159 (37.4)	63 (31.5)	1.000 (reference)			39 (27.9)	1.000 (reference)			17 (32.7)	1.000 (reference)		
TC	204 (48.0)	105 (52.5)	1.446 (0.970–2.155)	0.070	0.245	71 (50.7)	1.500 (0.943–2.386)	0.087	0.235	28 (53.8)	1.237 (0.646–2.369)	0.520	0.735
CC	62 (14.6)	32 (16.0)	1.543 (0.866–2.751)	0.141	0.212	30 (21.4)	2.238 (1.225–4.087)	0.009	0.045	7 (13.5)	1.152 (0.449–2.956)	0.769	0.869
Dominant			1.439 (0.982–2.109)	0.062	0.217		1.646 (1.061–2.553)	0.026	0.154		1.194 (0.642–2.219)	0.575	0.813
Recessive			1.115 (0.681–1.828)	0.665	0.798		1.694 (1.015–2.826)	0.044	0.190		0.925 (0.394–2.169)	0.857	0.912
*PAI-1* 12068 [rs1050955] G>A													
GG	163 (38.4)	61 (30.5)	1.000 (reference)			44 (31.4)	1.000 (reference)			18 (34.6)	1.000 (reference)		
GA	210 (49.4)	107 (53.5)	1.793 (1.187–2.707)	0.006	0.042	72 (51.4)	1.416 (0.898–2.233)	0.134	0.235	25 (48.1)	1.133 (0.588–2.184)	0.710	0.735
AA	52 (12.2)	32 (16.0)	1.946 (1.080–3.505)	0.027	0.162	24 (17.1)	1.795 (0.965–3.341)	0.065	0.142	9 (17.3)	1.647 (0.687–3.950)	0.263	0.658
Dominant			1.792 (1.212–2.648)	0.003	0.021		1.509 (0.982–2.319)	0.060	0.154		1.242 (0.672–2.296)	0.489	0.813
Recessive			1.314 (0.791–2.183)	0.292	0.458		1.472 (0.845–2.566)	0.173	0.288		1.505 (0.685–3.306)	0.309	0.773

AOR, adjusted odds ratio; 95% CI, 95% confidence interval; LAD, large-artery disease; SVD, small-vessel disease; CE, cardioembolism; N/A, not applicable; FDR, false discovery rate. ^a^ Adjusted for age, sex, hypertension, diabetes mellitus, hyperlipidemia, and smoking.

**Table 4 ijms-24-08041-t004:** Analysis of PAI-1 haplotypes in ischemic stroke patients, metabolic syndrome patients, and controls.

Characteristic	Stroke Controls (2n = 850)	Stroke Patients (2n = 1148)	OR (95% CI)	*p* ^a^	*FDR-P*
*PAI-1* −844G>A/−675 4G>5G/43G>A/9785G>A/10692T>C/11053T>G/12068G>A					
G-4G-G-G-T-T-G	18 (2.1)	16 (1.4)	1.000 (reference)		
A-4G-A-G-T-G-G	22 (2.6)	3 (0.3)	0.153 (0.039–0.611)	0.005	0.075
A-5G-G-G-C-T-A	0 (0.0)	37 (3.2)	84.090 (4.774–1481.000)	<0.0001	0.003
*PAI-1* −844G>A/−675 4G>5G/43G>A/10692T>C/11053T>G/12068G>A					
G-4G-G-T-T-G	21 (2.5)	19 (1.7)	1.000 (reference)		
G-4G-G-T-G-G	77 (9.0)	158 (13.8)	2.268 (1.151–4.467)	0.020	0.107
G-5G-A-C-T-A	23 (2.7)	48 (4.2)	2.307 (1.041–5.109)	0.045	0.181
A-4G-A-T-G-G	23 (2.7)	4 (0.3)	0.192 (0.056–0.658)	0.008	0.096
A-5G-G-C-T-A	0 (0.0)	39 (3.4)	87.100 (5.005–1516.000)	<0.0001	0.003
*PAI-1* −844G>A/−675 4G>5G/10692T>C/11053T>G/12068G>A					
G-4G-T-T-G	29 (3.4)	20 (1.8)	1.000 (reference)		
G-4G-T-G-G	87 (10.2)	155 (13.5)	2.583 (1.379–4.838)	0.004	0.031
G-4G-T-G-A	3 (0.4)	12 (1.0)	5.800 (1.448–23.240)	0.016	0.065
G-5G-C-T-A	158 (18.6)	217 (18.9)	1.991 (1.087–3.649)	0.032	0.105
A-4G-T-T-A	5 (0.6)	12 (1.1)	3.480 (1.060–11.430)	0.049	0.12
A-4G-C-T-A	17 (2.1)	35 (3.1)	2.985 (1.324–6.729)	0.010	0.058
A-5G-T-G-G	1 (0.2)	12 (1.0)	17.400 (2.092–144.800)	0.001	0.012
A-5G-C-T-A	0 (0.0)	43 (3.8)	125.200 (7.280–2153.000)	<0.0001	0.002
*PAI-1* −844G>A/−675 4G>5G/43G>A/10692T>C					
G-4G-G-T	112 (13.2)	198 (17.2)	1.000 (reference)		
G-4G-G-C	82 (9.6)	94 (8.2)	0.648 (0.445–0.945)	0.027	0.056
G-4G-A-T	15 (1.7)	4 (0.3)	0.151 (0.049–0.466)	0.0003	0.002
G-4G-A-C	11 (1.3)	6 (0.5)	0.309 (0.111–0.857)	0.022	0.056
G-5G-G-T	77 (9.1)	86 (7.5)	0.632 (0.430–0.929)	0.023	0.056
A-4G-G-T	281 (33.1)	331 (28.8)	0.666 (0.503–0.883)	0.005	0.019
A-4G-A-T	28 (3.3)	7 (0.6)	0.141 (0.060–0.334)	<0.0001	0.001
A-5G-G-C	0 (0.0)	40 (3.5)	45.910 (2.794–754.300)	<0.0001	0.001
*PAI-1* −844G>A/−675 4G>5G/10692T>C					
G-4G-T	127 (15.0)	201 (17.6)	1.000 (reference)		
A-4G-T	309 (36.4)	339 (29.5)	0.693 (0.529–0.909)	0.008	0.028
A-5G-T	3 (0.3)	19 (1.6)	4.002 (1.160–13.800)	0.021	0.049
A-5G-C	0 (0.0)	49 (4.3)	62.640 (3.827–1025.000)	<0.0001	0.0007
*PAI-1* −844G>A/−675 4G>5G					
G-4G	220 (25.9)	304 (26.5)	1.000 (reference)		
A-5G	3 (0.3)	70 (6.1)	16.890 (5.247–54.340)	<0.0001	0.0003

MDR, multifactor dimensionality reduction. ^a^ Fisher’s exact test. FDR: false discovery rate.

**Table 5 ijms-24-08041-t005:** Combined genotype analysis for *PAI-1* in metabolic syndrome and stroke patients and controls.

Genotype	MetS	MetS	OR (95% CI)	*p*		*FDR-P*	Stroke	Stroke	AOR (95% CI)	*p* ^a,b^	*FDR-P*
	Controls	Patients					Controls	Patients			
	(n = 764)	(n = 235)					(n = 425)	(n = 574)			
*PAI-1* −844G>A/−675 4G>5G (rs2227631/rs1799889)											
GG-4G4G	63 (8.2)	14 (6.0)	1.000 (reference)				38 (8.9)	39 (6.8)	1.000 (reference)		
GA-5G5G	12 (1.6)	8 (3.4)	3.000 (1.034–8.709)	0.043		0.172	2 (0.5)	18 (3.1)	8.389 (1.671–42.110)	0.01	0.030
*PAI-1* −844G>A/10692T> C (rs2227631/rs11178)											
GG-TT	70 (9.2)	21 (8.9)	1.000 (reference)				37 (8.7)	54 (9.4)	1.000 (reference)		
GA-CC	17 (2.2)	9 (3.8)	1.765 (0.687–4.535)		0.238	0.476	4 (0.9)	22 (3.8)	4.035 (1.145–14.217)	0.03	0.030
AA-TT	79 (10.3)	21 (8.9)	0.886 (0.447–1.758)		0.729	0.933	58 (13.6)	42 (7.3)	0.455 (0.238–0.872)	0.018	0.030
AA-TC	38 (5.0)	11 (4.7)	0.965 (0.421–2.212)		0.933	0.933	14 (3.3)	35 (6.1)	2.659 (1.126–6.276)	0.026	0.030
*PAI-1* −844G>A/11053T>G (rs2227631/rs7242)											
GG-TT	147 (19.2)	41 (17.4)	1.000 (reference)				83 (19.5)	105 (18.3)	1.000 (reference)		
AA-TT	9 (1.2)	5 (2.1)	1.992 (0.633–6.270)	0.239		0.432	1 (0.2)	13 (2.3)	10.968 (1.351–89.059)	0.025	0.030
*PAI-1* −675 4G>5G/43G>A (rs1799889/rs6092)											
4G4G-GG	258 (33.8)	94 (40.0)	1.000 (reference)				144 (33.9)	208 (36.2)	1.000 (reference)		
4G4G-GA	44 (5.8)	2 (0.9)	0.125 (0.030–0.525)	0.005		0.005	37 (8.7)	9 (1.6)	0.194 (0.088–0.428)	0.0001	0.0006
*PAI-1* −675 4G>5G/10692T>C (rs1799889/rs11178)											
4G4G-TT	174 (22.8)	64 (27.2)	1.000 (reference)				109 (25.6)	129 (22.5)	1.000 (reference)		
5G5G-TC	43 (5.6)	11 (4.7)	0.696 (0.338–1.431)	0.324		0.432	18 (4.2)	36 (6.3)	1.983 (1.011–3.891)	0.047	0.049
5G5G-CC	65 (8.5)	21 (8.9)	0.878 (0.497–1.552)	0.655		0.655	29 (6.8)	57 (9.9)	1.803 (1.030–3.154)	0.039	0.049
*PAI-1* −675 4G>5G/12068G>A (rs1799889/rs1050955)											
4G4G-GG	162 (21.2)	61 (26.0)	1.000 (reference)				101 (23.8)	122 (21.3)	1.000 (reference)		
5G5G-AA	47 (6.2)	14 (6.0)	0.791 (0.407–1.539)	0.490		0.490	20 (4.7)	41 (7.1)	1.908 (1.002–3.631)	0.049	0.049
*PAI-1* 43G>A/10692T>C (rs6092/rs11178)											
GG-TT	219 (28.7)	85 (36.2)	1.000 (reference)				130 (30.6)	174 (30.3)	1.000 (reference)		
GA-TT	32 (4.2)	7 (3.0)	0.564 (0.240–1.326)	0.189		0.252	29 (6.8)	10 (1.7)	0.269 (0.120–0.604)	0.001	0.004
*PAI-1* 43G>A/11053T>G (rs6092/rs7242)											
GG-TT	166 (21.7)	53 (22.6)	1.000 (reference)				87 (20.5)	132 (23.0)	1.000 (reference)		
GA-GG	20 (2.6)	1 (0.4)	0.157 (0.021–1.195)	0.074		0.232	16 (3.8)	5 (0.9)	0.247 (0.085–0.712)	0.01	0.015
*PAI-1* 43G>A/12068G>A (rs6092/rs1050955)											
GG-GG	232 (30.4)	82 (34.9)	1.000 (reference)				134 (31.5)	180 (31.4)	1.000 (reference)		
GA-GG	31 (4.1)	5 (2.1)	0.456 (0.172–1.213)	0.116		0.232	28 (6.6)	8 (1.4)	0.215 (0.092–0.502)	0.0003	0.0009
GA-AA	20 (2.6)	9 (3.8)	1.273 (0.557–2.908)	0.567		0.567	6 (1.4)	23 (4.0)	3.184 (1.195–8.483)	0.021	0.021
*PAI-1* 9785G>A/12068G>A (rs2227694/rs1050955)											
GG-GG	237 (31.0)	78 (33.2)	1.000 (reference)				151 (35.5)	164 (28.6)	1.000 (reference)		
GG-GA	357 (46.7)	104 (44.3)	0.885 (0.632–1.239)	0.477		0.636	195 (45.9)	266 (46.3)	1.415 (1.041–1.923)	0.027	0.035
GG-AA	114 (14.9)	37 (15.7)	0.986 (0.629–1.548)	0.952		0.952	50 (11.8)	101 (17.6)	1.944 (1.270–2.974)	0.002	0.008
*PAI-1* 10692T>C/12068G>A (rs11178/rs1050955)											
TC-GA	283 (37.0)	79 (33.6)	0.787 (0.540–1.145)	0.211		0.636	150 (35.3)	212 (36.9)	1.485 (1.051–2.098)	0.025	0.035
CC-AA	81 (10.6)	23 (9.8)	0.800 (0.466–1.375)	0.420		0.636	38 (8.9)	66 (11.5)	1.707 (1.039–2.804)	0.035	0.035

Results that lacked statistical significance (e.g., *p* > 0.05) were excluded. OR, odds ratio; AOR, adjusted odds ratio; MetS, metabolic syndrome; 95% CI, 95% confidence interval; FDR, false discovery rate; ^a^ Fisher’s exact test. ^b^ Adjusted for age, sex, hypertension, diabetes mellitus, hyperlipidemia, and smoking.

## Data Availability

All supporting data of the study are available from the corresponding authors upon request.

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
