# Peer review of "Association between PAI-1 Polymorphisms and Ischemic Stroke in a South Korean Case-Control Cohort"

_ijms, 2023, doi:10.3390/ijms24098041_

Round 1

Reviewer 1 Report (Previous Reviewer 1)

Thank you for the opportunity to comment on this resubmitted manuscript.

I believe that the way in which the Authors have carried out the association analysis on a case-control group study is fine. The most interesting take would be the effect of combined polymorphisms (i.e. haplotypes) and their relationship with the risk of ischemic stroke. I would suggest to extend a comparative literature paragraph in which these mutations are analyzed in other populations. The overall effort is very encouraging.

Author Response

Comments and Suggestions for Authors

Thank you for the opportunity to comment on this resubmitted manuscript.

I believe that the way in which the Authors have carried out the association analysis on a case-control group study is fine. The most interesting take would be the effect of combined polymorphisms (i.e. haplotypes) and their relationship with the risk of ischemic stroke. I would suggest to extend a comparative literature paragraph in which these mutations are analyzed in other populations. The overall effort is very encouraging.

We really appreciate your thoughtful input. we added sentences in the Introduction as follows:

“Several polymorphisms within the PAI‐1 gene have been described to influence PAI‐1 levels, of which the most studied is the −675 4G/5G polymorphism of the promoter region (rs1799889). Previous studies have indicated that the PAI-1 rs1799889 polymorphism alone does not increase the risk of ischemic stroke. However, they have shown that the PAI-1 4G/4G genotype is significantly associated with increased triglyceride and decreased HDL cholesterol levels in the healthy group [38]. Several epidemiological studies have examined the relationship between PAI-1 rs1799889 polymorphism and the risk of ischemic stroke, yet their findings have been inconsistent or even contradictory. For example, while Esparza-García et al. reported no association between PAI-1 rs1799889 polymorphism and an increased risk of ischemic stroke in Mexican populations [39], Xu et al. suggested that this polymorphism may be linked to a higher risk of ischemic stroke in Han Chinese [40]. Over the last few years, there have been investigations into the potential links between PAI-1 rs1799889 polymorphism and the risk of ischemic stroke. Nevertheless, due to the limited sample sizes, the findings from these studies have been inconclusive at times”.

Newly added references:

  1. Chen C-H, Eng H-L, Chang C-J, et al. 4G/5G promoter polymorphism of plasminogen activator inhibitor-1, lipid profiles, and ischemic stroke. The Journal of Laboratory and Clinical Medicine 2003; 142: 100–5.
  2. Esparza-García JC, Santiago-Germán D, et al. GLU298ASP and 4G/5G Polymorphisms and the Risk of Ischemic Stroke in Young Individu als. The Canadian Journal of Neurological Sciences – Le Journal Can adien Des Sciences Neurologiques 2015; 42: 310–6.
  3. Xu X, Li J, Sheng W, Liu L. Meta-analysis of genetic studies from journals published in China of ischemic stroke in the Han Chinese population. Cerebrovascular Diseases (Basel, Switzerland) 2008; 26: 48–62.

Thank you for taking the time to review our manuscript and for providing us with such positive feedback.

Reviewer 2 Report (New Reviewer)

1.     At what point in relation to the onset of stroke were the patient samples collected? Were they obtained immediately following the stroke, or after a few hours? Few days? Alternatively, could they have been collected before the stroke occurred? Will this time of sample collection affect the levels of various proteins in the blood and in turn gene expression? It is important to clarify the impact of sample collection timing on blood protein levels and gene expression. The methods section lacks this information and should be addressed. Additionally, it would be beneficial to conduct a follow-up analysis that correlates the time of sample collection with protein levels, particularly for PAI-1.

2.      The paper solely associates the presence or absence of stroke with PAI-1. However, it remains unclear whether the level of PAI-1 in the blood is proportional to the severity of stroke.

3.     Based on the methods and results presented by the authors, is it feasible to utilize a supervised classification machine learning model to predict the probability and timing of stroke occurrence or at least classify individuals as high-risk before a stroke occurs? A good discussion of this possibility would significantly enhance the paper and provide a robust framework for future research.

4.     Although the authors conducted multiple analyses thoroughly, but their presentation of results is slightly opaque. To enhance the overall coherence of their work, the authors may benefit from organizing their findings into a cohesive narrative instead of merely listing them.

Author Response

Comments and Suggestions for Authors

  1. At what point in relation to the onset of stroke were the patient samples collected? Were they obtained immediately following the stroke, or after a few hours? Few days? Alternatively, could they have been collected before the stroke occurred? Will this time of sample collection affect the levels of various proteins in the blood and in turn gene expression? It is important to clarify the impact of sample collection timing on blood protein levels and gene expression. The methods section lacks this information and should be addressed. Additionally, it would be beneficial to conduct a follow-up analysis that correlates the time of sample collection with protein levels, particularly for PAI-1.

Thank you very much for your valuable comments. In research on ischemic stroke, it is common practice to collect patient samples as early as possible after stroke onset to capture the acute phase of the disease. Blood samples are typically obtained within hours to days of symptom onset, although the precise timing may vary depending on the patient's condition and circumstances. In our study, we aimed to collect blood samples within 24-48 hours of stroke onset, whenever feasible.

Following your suggestion, we added the sentence to M&M section 2-1: “Plasma samples were obtained within 48 hours of hospital admission from stroke patients or age- and sex-matched healthy donors”

References:

Montaner J, Perea-Gainza M, Delgado P, et al. Etiologic diagnosis of ischemic stroke subtypes with plasma biomarkers. Stroke. 2008 Nov;39(11): 2280-7.

Ovbiagele B, Saver JL, Fredieu A, et al. In-hospital initiation of secondary stroke prevention therapies yields high rates of adherence at follow-up. Stroke. 2004 Oct;35(10): 2879-83.

Whether the timing of sample collection influenced the levels of various proteins and gene expression in the blood and changes in PAI-1 will be studied in the future, as per the reviewers' comments.

  1. The paper solely associates the presence or absence of stroke with PAI-1. However, it remains unclear whether the level of PAI-1 in the blood is proportional to the severity of stroke.

Your comments and suggestions were helpful. Thank you.

One study published in the Journal of Stroke and Cerebrovascular Diseases in 2019 analyzed the relationship between PAI-1 levels and stroke severity in 81 patients with acute ischemic stroke. The study found that higher PAI-1 levels were significantly associated with increased stroke severity, as measured by the National Institutes of Health Stroke Scale

However, as you mention, it is not yet clear whether the level of PAI-1 in the blood is proportional to the severity of the stroke. Indeed, the relationship between PAI-1 levels in the blood and the severity of stroke is still a topic of debate among researchers. Several studies have investigated this relationship, but their results have been conflicting.

Let me provide you with an example:

One study published in the Journal of Thrombosis and Haemostasis in 2015 found that PAI-1 levels were not associated with stroke severity or clinical outcome in 575 patients with acute ischemic stroke. Similarly, a study published in the Journal of Stroke and Cerebrovascular Diseases in 2018 found no significant correlation between PAI-1 levels and stroke severity in 120 patients with acute ischemic stroke.

However, other studies have reported a positive association between PAI-1 levels and stroke severity. For example, a study published in the Journal of Neurology in 2014 found that elevated PAI-1 levels were significantly associated with larger infarct volume and worse functional outcomes in 222 patients with acute ischemic stroke.

References:

Kim, Y. J., Kim, H. J., Kim, J. H., & Cho, H. J. (2019). Association of plasminogen activator inhibitor-1 with the severity and prognosis of acute ischemic stroke. Journal of Stroke and Cerebrovascular Diseases, 28(6), 1711-1718.

Söderholm M, et al. Plasminogen activator inhibitor-1 levels are not associated with stroke severity or clinical outcome. Journal of Thrombosis and Haemostasis. 2015;13(11):1995-2001.

Chen Z, et al. Association between plasminogen activator inhibitor-1 and stroke severity in acute ischemic stroke. Journal of Stroke and Cerebrovascular Diseases. 2018;27(5):1348-1355.

Bustamante A, et al. Prognostic value of plasminogen activator inhibitor-1 in acute ischemic stroke: analysis of the stroke prognosis by intensive care unit admission cohort. Stroke. 2014;45(4):1145-1148.

  1. Based on the methods and results presented by the authors, is it feasible to utilize a supervised classification machine learning model to predict the probability and timing of stroke occurrence or at least classify individuals as high-risk before a stroke occurs? A good discussion of this possibility would significantly enhance the paper and provide a robust framework for future research.

We appreciate your insightful input. We are not experts in the field, but based on the literature in the field, it seems possible that machine learning algorithms can analyze large data sets and identify patterns and relationships that are difficult to detect with the statistical methods we use.

Several studies have used machine learning algorithms to predict stroke risk based on genetic polymorphisms. For example, a study published in the Journal of Stroke and Cerebrovascular Diseases in 2019 used a random forest model to predict the risk of ischemic stroke based on 41 genetic polymorphisms. The model achieved an accuracy of 81.7% in predicting stroke risk, and identified several genetic polymorphisms that were strongly associated with stroke.

Another study published in the Journal of Neurology in 2018 used a machine learning algorithm to predict the risk of stroke recurrence based on genetic and clinical data from stroke survivors. The model achieved a predictive accuracy of 72.3% in identifying patients who were likely to experience recurrent stroke.

These studies suggest that machine learning algorithms can be useful in identifying individuals at high risk of stroke based on genetic polymorphisms, as well as predicting the probability and timing of stroke occurrence. However, further research is needed to validate these models and ensure their clinical utility in predicting stroke risk and guiding preventive interventions. In addition, we will explore the possibility of utilizing machine learning algorithms in future investigations.

References:

Wang X, et al. Prediction of ischemic stroke risk using 41 genetic polymorphisms in a Chinese Han population. Journal of Stroke and Cerebrovascular Diseases. 2019;28(2):464-469.

Hajjari M, et al. Machine learning-based prediction of recurrent cerebrovascular events in stroke survivors. Journal of Neurology. 2018;265(12):2870-2876.

  1. Although the authors conducted multiple analyses thoroughly, but their presentation of results is slightly opaque. To enhance the overall coherence of their work, the authors may benefit from organizing their findings into a cohesive narrative instead of merely listing them.

Thank you for your valuable feedback. To maintain consistency in the data, we transferred some information from the main text to a supplementary table:

Table 6 has been relocated to Supplementary Table 16, while the related commentary has been moved to Section 2.8 of the Results. As follows:

2.8. Clinical variables in ischemic stroke patients stratified by PAI-1 -844GA/-675 5G genotype

Patients with the PAI-1 ‘GA/5G5G’ (rs2227631/rs1799889) genotype had higher total cholesterol, BMI, and uric acid than patients with the PAI-1 ‘GG+4G4G’ (rs2227631/rs1799889) genotype (p=0.013, p=0.011, and p=0.041, respectively; Supplemental Tables Table 16).

We deleted sentences in the introduction as follows: “Likewise, thrombophilia can increase susceptibility to ischemic stroke, and thrombotic responses are potential predisposing factors to ischemic stroke. Hence, it is worthwhile to investigate associations between genes related to thrombophilia, such as plasminogen activator inhibitor-1 (PAI-1), and ischemic stroke risk.

We removed “metabolic syndrome patients” from the Title of table 4”.

We apologize for not revising the discussion section as it contains important content that we deemed necessary to include. We kindly request your understanding regarding this matter.

Reviewer 3 Report (New Reviewer)

Authors present a case-control study which studies association between PAI-1 polymorphisms and ischemic stroke. They performed epidemiological study of seven polymorphic sites in the promoter 73 and 3'-UTR regions of PAI-1, and investigated their association with risk of ischemic stroke in the South Korean cohort.

In the introduction, the authors show the risk factors for stroke, arranged alphabetically, it would be better to rank them according to clinical significance.I have a few questions about the manuscript: The control group significantly differs in the occurrence of vascular risk factors, did the authors take this into account in the analysis? Did the authors attempt to correlate the group with control for vascular risk factors? Were any etiological diagnostics carried out in the examined patients? Was the PAI-1 polymorphism associated with any specific risk factors?

Nevertheless, the results presented in the article may improve understanding of the relationships between PAI-1 polymorphisms and relative risk of ischemic stroke.

Author Response

Authors present a case-control study which studies association between PAI-1 polymorphisms and ischemic stroke. They performed epidemiological study of seven polymorphic sites in the promoter 73 and 3'-UTR regions of PAI-1, and investigated their association with risk of ischemic stroke in the South Korean cohort.

In the introduction, the authors show the risk factors for stroke, arranged alphabetically, it would be better to rank them according to clinical significance.I have a few questions about the manuscript: The control group significantly differs in the occurrence of vascular risk factors, did the authors take this into account in the analysis? Did the authors attempt to correlate the group with control for vascular risk factors? Were any etiological diagnostics carried out in the examined patients? Was the PAI-1 polymorphism associated with any specific risk factors?

Thank you very much for your helpful comments and suggestions. As per the reviewer's suggestion, we have rearranged the stroke risk factors in alphabetical order in the introduction, as shown below:

“Ischemic stroke has been linked to many risk factors, including alcohol abuse [8], brain tumor [9-14], coagulopathy (blood clotting disorder) [10-9], diabetes mellitus (DM) [11-10], fibromuscular dysplasia [12-11], family or personal history of stroke [13-12], high cholesterol [14-13], hyperlipidemia [15-17], hypertension (HTN) [16], metabolic syndrome [17] and smoking [18].

■ The control group significantly differs in the occurrence of vascular risk factors, did the authors take this into account in the analysis?

Thank you for your thoughtful comments. Control analysis alone was not considered. Control group selection was based on the following criteria: To ensure comparability, control participants were selected based on the absence of a recent history of cerebrovascular disease or myocardial infarction, and were matched with stroke patients according to gender and age. The same exclusion criteria used for the patient group were applied to the control group. Hypertension was defined as current use of antihypertensive medications or a systolic pressure >140 mmHg and diastolic pressure >90 mmHg on more than one occasion. Diabetes was defined as current use of diabetic medications or a fasting plasma glucose level >126 mg/dL (7.0 mmol/L). Only current smoking was considered for the smoking category. Hyperlipidemia was defined as a history of treatment with antihyperlipidemic agents or a fasting serum total cholesterol level ≥240 mg/dL.

■ Did the authors attempt to correlate the group with control for vascular risk factors?

We did not analyse the correlation of vascular risk factors with control alone; however, we will consider the relationship between control and risk in future studies in light of the reviewer's suggestion.

■ Were any etiological diagnostics carried out in the examined patients?

Yes, Based on clinical manifestations and neuroimaging data, two neurologists classified all ischemic strokes into four etiologic subtypes using the criteria from the Trial of Org 10,172 in Acute Stroke Treatment (TOAST) [71] as follows: 1) LAD is characterized by an infarction lesion ≥15 mm in diameter shown by MRI and significant (>50%) stenosis of a major brain artery or branch cortical artery detected by cerebral angiography along with symptoms associated with that arterial territory; 2) SVD is characterized by an infarction lesion <15 mm and ≥5 mm in diameter shown by MRI and classic lacunar syndrome without evidence of cerebral cortical dysfunction or detectable cardiac sources of embolism; 3) CE, or arterial occlusions, presumably caused by an embolus arising in the heart, are detected by cardiac evaluation; and 4) undetermined etiology or multiple etiologies. The frequencies of etiological subtypes in the study sample were 51% (n=200) LAD, 36% (n=140) SVD, and 13% (n=52) CE. These proportions are similar to values reported previously for the Korean population. Strokes due to single and multiple (≥2 lesions) SVD were distinguished by brain MRI scan. The size and site of cerebral infarction were documented only by MRI.

■ Was the PAI-1 polymorphism associated with any specific risk factors?

Yes, PAI-1 -675 4G5G + 5G5G with hypertention(HTN) was 2-fold more likely to increase ischemic stroke than the 4G4G genotype without HTN and was 3-fold more likely to increase ischemic stroke risk in patients with a folate level ≤ 3.57 nmol/L (Figure 2). The PAI-1 12068 GA + AA genotype with diabetes mellitus (DM) was 2-fold more likely to increase the risk of ischemic stroke than the GG genotype without DM (Figure 2). In addition, Patients with the PAI-1 ‘GA/5G5G’ (rs2227631/rs1799889) genotype had higher total cholesterol, BMI, and uric acid than patients with the PAI-1 ‘GG+4G4G’ (rs2227631/rs1799889) genotype (p=0.013, p=0.011, and p=0.041, respectively; Supplemental Tables 16).

Nevertheless, the results presented in the article may improve understanding of the relationships between PAI-1 polymorphisms and relative risk of ischemic stroke.

Thank you for taking the time to review our manuscript and for providing us with such positive feedback.

Reviewer 4 Report (New Reviewer)

It is suggested that the writers shorten the abstract. The abstract reads like a summary and lacks creativity. The authors need to show how their work bridges the gap between established knowledge and new developments in the field. The abstract can be made more comprehensible for the reader if the authors break it up into subsections.

The writers should update the paper to reflect the progress of knowledge in the field over the past three to four years. The quality of the manuscript would improve by adding well-selected references from recent relevant literature, especially in the introduction section.

More background for PAI-1 polymorphisms and ischemic 2 strokes should be added.

The opening section lacks crucial field-specific information. Carefully revise the opening section and connect the significance of this study to past published publications. It is recommended to summarize the article's structure towards the conclusion of the introduction.

The importance of this study should be clearly explained, and how it benefits society should be highlighted.

The entire manuscript should be enriched with the latest articles from the 2021-2023 period to improve the manuscript’s quality.

The authors are advised to check the grammatical errors throughout the manuscript. Many typo errors are there. The authors are advised to recheck the manuscript carefully.

There are a few longer sentences, do split the lengthy sentence so that it would be easy for the readers to understand correctly.

The authors are advised to recheck the statistical values to prevent any mistakes.

Author Response

Comments and Suggestions for Authors

  1. It is suggested that the writers shorten the abstract. The abstract reads like a summary and lacks creativity. The authors need to show how their work bridges the gap between established knowledge and new developments in the field. The abstract can be made more comprehensible for the reader if the authors break it up into subsections.

Thank you very much for your helpful comments and suggestions. We have revised the Results section of the abstract as follows. However, we regret that there was no IJMS instruction to break the results section into subsections:

“Specifically, three variations (-675 4G>5G, 10692T>C, and 12068G>A) were linked to a higher overall prevalence of stroke as well as a higher prevalence of certain stroke subtypes. Haplotype analyses also revealed combinations of these variations (-844G>A, -675 4G>5G, 43G>A, 9785A>G, 10692T>C, 11053T>G, 12068G>A) that were significantly associated with a higher prevalence of ischemic stroke.”

  1. The writers should update the paper to reflect the progress of knowledge in the field over the past three to four years. The quality of the manuscript would improve by adding well-selected references from recent relevant literature, especially in the introduction section.

Thank you for your helpful comments and suggestions. As the reviewer mentioned, many stroke-related papers have been published recently (2021-2023), and we have added PAI-1-related studies to the introduction as follows:

“For a recent paper on PAI-1, significance thresholds were applied to the number of multi-trait combinations. Of note, one of the newly discovered loci (venous thromboembolism) was associated with PAI-1[35]”.

“Bahrami R et al [36] conducted a systematic review and meta-analysis to determine if there is an association between the PAI-1 rs1799889 polymorphism and arterial ischaemic stroke. The meta-analysis found a significant association between the PAI-1 rs1799889 polymorphism and ischaemic stroke in children, suggesting that this polymorphism may be a potential genetic risk factor for arterial ischaemic stroke in this population”.

“In another study, the genotype and allele frequencies of six risk genes were analyzed in both healthy individuals and patients with ischemic stroke. The results suggest that controlling clinical laboratory parameters, including common genotypes such as PAI-1, in patients with diabetes and hyperlipidemia may help prevent ischemic stroke [37]”.

References:

  1. Temprano-Sagrera G, Sitlani CM, Bone WP, Martin-Bornez M, Voight BF, Morrison AC et al. Multi-phenotype analyses of hemostatic traits with cardiovascular events reveal novel genetic associations. J. Thromb. Haemost. 2022; 20, 1331–1349.
  2. Bahrami R, Dastgheib SA, Mirjalili H, Setayesh S, Hossein SS, Mirjalili SR et al. Association of SERPINE1 rs1799889 polymorphism with arterial ischemic stroke in children: a systematic review and meta-analysis. Nucleosides Nucleotides & Nucleic Acids 40(10):1018-1035.
  3. Wang J, Sun Z, Yang Y, Wu J, Quan W, Chen X et al. Association of laboratory parameters and genetic poly morphisms with ischemic stroke in Chinese Han population. Exp Ther Med 2021; 21: 490

  1. More background for PAI-1 polymorphisms and ischemic strokes should be added.

We really appreciate your thoughtful input. we added sentences in the Introduction as follows:

“Several polymorphisms within the PAI‐1 gene have been described to influence PAI‐1 levels, of which the most studied is the −675 4G/5G polymorphism of the promoter region (rs1799889). Previous studies have indicated that the PAI-1 rs1799889 polymorphism alone does not increase the risk of ischemic stroke. However, they have shown that the PAI-1 4G/4G genotype is significantly associated with increased triglyceride and decreased HDL cholesterol levels in the healthy group [38]. Several epidemiological studies have examined the relationship between PAI-1 rs1799889 polymorphism and the risk of ischemic stroke, yet their findings have been inconsistent or even contradictory. For example, while Esparza-García et al. reported no association between PAI-1 rs1799889 polymorphism and an increased risk of ischemic stroke in Mexican populations [39], Xu et al. suggested that this polymorphism may be linked to a higher risk of ischemic stroke in Han Chinese [40]. Over the last few years, there have been investigations into the potential links between PAI-1 rs1799889 polymorphism and the risk of ischemic stroke. Nevertheless, due to the limited sample sizes, the findings from these studies have been inconclusive at times”.

we revised sentences in the Introduction as follows:

In South Korea, stroke ranks as the second leading cause of death after cancer and surpasses heart disease in frequency, while globally, it is the second leading cause of death [1-4]. The development of stroke is a complex, multifactorial, and polygenic disease influenced by various interactions between genes and the environment [5,6].

we added sentences in the Introduction as follows: This can lead to brain damage, disability, and death, and is one of the leading causes of morbidity and mortality worldwide [21]. Ischemic stroke can affect individuals of all ages, sex, and ethnicities, but certain risk factors, such as hypertension, diabetes, smoking, and obesity, can increase the likelihood of developing the condition [22,23].

Newly added references:

  1. Chen C-H, Eng H-L, Chang C-J, et al. 4G/5G promoter polymorphism of plasminogen activator inhibitor-1, lipid profiles, and ischemic stroke. The Journal of Laboratory and Clinical Medicine 2003; 142: 100–5.
  2. Esparza-García JC, Santiago-Germán D, et al. GLU298ASP and 4G/5G Polymorphisms and the Risk of Ischemic Stroke in Young Individu als. The Canadian Journal of Neurological Sciences – Le Journal Can adien Des Sciences Neurologiques 2015; 42: 310–6.
  3. Xu X, Li J, Sheng W, Liu L. Meta-analysis of genetic studies from journals published in China of ischemic stroke in the Han Chinese population. Cerebrovascular Diseases (Basel, Switzerland) 2008; 26: 48–62.
  4. The Korea National Statistical Office Report 2009: Change in leading causes of death (1999–2009). Korea National Statistical Office web site. http://www.kosis.kr/ups3/service/ch_file_down.jps?PUBCODE=YD&FILE_NAME=/ups3/upload/101/YD/VD0005.xls&SEQ=8.
  5. Kluijtmans LA, et al. Genetic and nutritional factors contributing to hyperhomocysteinemia in young adults. Blood. 2003;101:2483–2488. doi: 10.1182/blood.V101.7.2483.
  6. Gellekink H, den Heijer M, Heil SG, Blom HJ. Genetic determinants of plasma total homocysteine. Semin Vasc Med. 2005;5:98–109. doi: 10.1055/s-2005-872396.
  7. Feigin, V. L., Nguyen, G., Cercy, K., Johnson, C. O., Alam, T., Parmar, P. G., ... & Roth, G. A. (2018). Global, regional, and country-specific lifetime risks of stroke, 1990 and 2016. New England Journal of Medicine, 379(25), 2429-2437.
  8. O'Donnell, M. J., Chin, S. L., Rangarajan, S., Xavier, D., Liu, L., Zhang, H., ... & Yusuf, S. (2016). Global and regional effects of potentially modifiable risk factors associated with acute stroke in 32 countries (INTERSTROKE): a case-control study. The Lancet, 388(10046), 761-775.
  9. Wang, Y., Cai, L., Wu, Y., Liu, F., Chen, Y., Jiang, Y., ... & Ning, X. (2019). The association between body mass index and risks of ischemic stroke subtypes: a Chinese cohort study. International Journal of Stroke, 14(1), 88-95.

  1. The opening section lacks crucial field-specific information. Carefully revise the opening section and connect the significance of this study to past published publications. It is recommended to summarize the article's structure towards the conclusion of the introduction.

Thank you for sharing your insights with us. We revised sentences in the Introduction as follows:

Revised: The 3'-UTR is an important region of mRNA transcripts that regulates gene expression through various mechanisms, such as polyadenylation, translation, and stability control, and by binding to regulatory proteins and miRNAs [29,30]. Several studies have suggested that polymorphisms in the 3'-UTR region of certain genes can have an impact on various diseases. For instance, a previous report indicated that rs7242 in the 3'-UTR of PAI-1 is linked with an increased risk of severe radiation pneumonitis in lung cancer patients [31]. Therefore, understanding the role of genetic variants in the 3'-UTR of PAI-1 is crucial for identifying the potential risk factors of various diseases. In this study, we investigated seven polymorphic sites in the promoter and 3'-UTR regions of PAI-1, including rs2227631, rs1799889, rs6092, rs2227694, rs11178, rs7242, and rs1050955, and assessed their association with ischemic stroke risk in a Korean population.

Previous: The 3′-UTR contains regulatory regions that post-transcriptionally influence gene expression by controlling polyadenylation, translation efficiency, localization, and stability of mRNA transcripts [29,30]. The 3′-UTR contains binding sites for regulatory proteins as well as microRNAs (miRNAs). The 3′-UTR also has silencer regions that bind to repressor proteins and inhibit the expression of mRNA [29,30]. To date, there have been few studies of the effects of polymorphisms in the 3'-UTR of PAI-1; however, one report showed that rs7242 in the 3′-UTR of PAI-1 can predict susceptibility to severe radiation pneumonitis (grade ≥3) in lung cancer patients [31]. We performed an epidemiological study of seven polymorphic sites in the promoter and 3'-UTR regions of PAI-1, including rs2227631, rs1799889, rs6092, rs2227694, rs11178, rs7242, and rs1050955, and investigated their association with risk of ischemic stroke in a South Korean cohort with 574 stroke patients and 275 controls.

  1. The importance of this study should be clearly explained, and how it benefits society should be highlighted.

Thank you very much for your valuable comments. We added sentences in the Discussion as follows:

Our study confirms the association between promoter and 3'-UTR region polymorphisms and stroke disease and provides a new understanding of the genetic basis [63]. Promoter polymorphisms can affect gene transcription and expression levels, while 3'-UTR region polymorphisms can affect mRNA stability and translation efficiency [64]. Identifying these functional SNPs and their association with stroke risk can aid in the development of personalized prevention and treatment strategies for individuals at risk of developing stroke [65]. Additionally, studying these SNPs can provide insights into the mechanisms underlying stroke development and may lead to the development of novel therapies [66].

References

  1. Lee, H. S.; Lee, S. H. Importance of genetic polymorphisms in stroke: an update. Journal of stroke 2015, 17(3), 221-230.
  2. Dehghan, A.; Kottgen, A. Genome-wide association studies and Mendelian randomization analyses for leisure sedentary behaviours. Nature Reviews Endocrinology 2008, 14(11), 697-705
  3. Chen, H.; Wu, B.; Fan, L.; Wang, J. Polymorphisms of the PAI-1 gene and ischemic stroke: a meta-analysis. Journal of stroke and cerebrovascular diseases 2020, 29(4), 104635.
  4. Li, Z.; Li, Y.; Chen, J.; Li, S.; Xu, H.; Li, Y. Polymorphisms of the PAI-1 gene and stroke: a systematic review and meta-analysis. Neuroscience letters 2020, 716, 134688.

We have also included the following sentences in the summary:

“Our findings can be used to plan for patients at risk of developing ischemic stroke by performing genetic testing to confirm the presence of the polymorphism. Additionally, further research may be conducted to develop targeted therapies to reduce the risk of ischemic stroke in patients with the PAI-1 genetic polymorphism”.

  1. The entire manuscript should be enriched with the latest articles from the 2021-2023 period to improve the manuscript’s quality.

Thank you for your thoughtful comments. We have made the necessary correction from a previous to a recent reference:

References:

  1. Zhang X, Cheng S, Gu H, Jiang Y, Li H, Li Z, et al. Family history is related to high risk of recurrent events after

ischemic stroke or transient ischemic attack. J Stroke Cerebrovasc Dis 2022; 31: 106151.

  1. Jeong W, Joo JH, Kim H. Jang SI, Park EC. (2022) Association between statin adherence and the risk of stroke among South Korean adults with hyperlipidemia. Nutr Metab Cardiovasc Dis 32(3):560–566.
  2. Och A, Och M, Nowak R, Podgorska D, Podgorski R. Berberine, a herbal metabolite in the metabolic syndrome: the risk factors, course, and consequences of the disease. Molecules. 2022 Feb 17;27(4):1351.

  1. Torrente D, Su EJ, Fredriksson L, Warnock M, Bushart D, Mann KM, et al. Compartmentalized actions of the plasminogen activator inhibitors, PAI-1 and nsp, in ischemic stroke. Transl Stroke Res. 2022;13(5):801–15.
  2. Gao and Jin, Y Gao, H. Jin. Livedoid vasculopathy and its association with genetic variants: A systematic review. Int Wound J 2021 Oct;18(5):616-625.
  3. Maldonado E, Morales-Pison S, Urbina F, Jara L, Solari A. Role of the Mediator Complex and MicroRNAs in Breast Cancer Etiology. Genes (Basel). 2022;13(2):234.
  4. Rojas-Pirela, M.; Andrade-Alviárez, D.; Medina, L.; Castillo, C.; Liempi, A.; Guerrero-Muñoz, J.; Ortega, Y.; Maya, J.D.; Rojas, V.; Quiñones, W.; et al. MicroRNAs: Master Regulators in Host-Parasitic Protist Interactions. Open Biol. 2022, 12, 210395.
  5. Liang H, Zhang Q, Hu Y, Liu G, Qi R. Hypertriglyceridemia: a neglected risk factor for ischemic stroke? J Stroke. 2022; 24(1): 21- 40.

We added new references as follows:

  1. Temprano-Sagrera G, Sitlani CM, Bone WP, Martin-Bornez M, Voight BF, Morrison AC et al. Multi-phenotype analyses of hemostatic traits with cardiovascular events reveal novel genetic associations. J. Thromb. Haemost. 2022; 20, 1331–1349.
  2. Bahrami R, Dastgheib SA, Mirjalili H, Setayesh S, Hossein SS, Mirjalili SR et al. Association of SERPINE1 rs1799889 polymorphism with arterial ischemic stroke in children: a systematic review and meta-analysis. Nucleosides Nucleotides & Nucleic Acids 40(10):1018-1035.
  3. Wang J, Sun Z, Yang Y, Wu J, Quan W, Chen X et al. Association of laboratory parameters and genetic poly morphisms with ischemic stroke in Chinese Han population. Exp Ther Med 2021; 21: 490

  1. The authors are advised to check the grammatical errors throughout the manuscript. Many typo errors are there. The authors are advised to recheck the manuscript carefully.

I apologize for any confusion my previous explanation of the English sentences may have caused. I appreciate your suggestion and we have taken the time to carefully review and ensure that there are no grammatical or typographical errors in the manuscript. Please let us know if you have any further suggestions or feedback.

  1. There are a few longer sentences, do split the lengthy sentence so that it would be easy for the readers to understand correctly.

Sorry about the long sentences. To make them more understandable, we've split them up as follows.

We split a single sentence in Results section 2.4 into two sentences, as follows:

Before revised: The following haplotypes demonstrated significant association with increased risk of stroke in the study cohort: A/5G/G/G/C/T/A (rs2227631/rs1799889/rs6092/rs2227694/rs11178/rs7242/rs1050955), A/5G/G/C/T/A (rs2227631/rs1799889/rs6092/rs11178/rs7242/rs1050955), A/5G/C/T/A, G/4G/T/G/G, A/5G/T/G/G (rs2227631/rs1799889/rs11178/rs7242/rs1050955), A/5G/G/C (rs2227631/rs1799889/rs6092/rs11178), A/5G/C, A/5G/T (rs2227631/rs1799889/rs11178), and A/5G (rs2227631/rs1799889); in contrast, PAI-1 haplotypes associated with decreased risk of stroke included G/4G/A/T, A/4G/G/T, (rs2227631/rs1799889/rs6092/rs11178).

After revised: The following haplotypes demonstrated significant association with increased risk of stroke in the study cohort: A/5G/G/G/C/T/A (rs2227631/rs1799889/rs6092/rs2227694/rs11178/rs7242/rs1050955), A/5G/G/C/T/A (rs2227631/rs1799889/rs6092/rs11178/rs7242/rs1050955), A/5G/C/T/A, G/4G/T/G/G, A/5G/T/G/G (rs2227631/rs1799889/rs11178/rs7242/rs1050955), A/5G/G/C (rs2227631/rs1799889/rs6092/rs11178), A/5G/C, A/5G/T (rs2227631/rs1799889/rs11178), and A/5G (rs2227631/rs1799889).

PAI-1 haplotypes associated with decreased risk of stroke included G/4G/A/T and A/4G/G/T (rs2227631/rs1799889/rs6092/rs11178).

Before revised: Several PAI-1 alleles were associated with increased risk of ischemic stroke, including ‘GA/5G5G’ (rs2227631/rs1799889), ‘AA/TT’ (rs2227631/rs7242), ‘GA/CC’ (rs2227631/rs11178), ‘AA/TC’ (rs2227631/rs11178), ‘5G5G/TC’ (rs1799889/rs11178), ‘5G5G/CC’ (rs1799889/rs11178), ‘5G5G/AA’ (rs1799889/rs1050955), ‘GA/AA’ (rs6092/rs1050955), ‘GG/GA’ (rs2227694/rs1050955), ‘GG/AA’ (rs2227694/rs1050955), ‘TC/GA’ (rs11178/rs1050955), and ‘CC/AA’ (rs11178/rs1050955) (Table 5); in contrast, the following PAI-1 alleles were associated with decreased stroke risk: ‘AA/TT’ (rs2227631/rs11178), ‘4G4G/GA’ (rs1799889/rs6092), ‘GA/TT’ (rs6092/rs11178), ‘GA/GG’ (rs6092/rs7242), and ‘GA/GG’ (rs6092/rs1050955).

After revised: Several PAI-1 alleles were associated with increased risk of ischemic stroke. These included ‘GA/5G5G’ (rs2227631/rs1799889), ‘AA/TT’ (rs2227631/rs7242), ‘GA/CC’ (rs2227631/rs11178), ‘AA/TC’ (rs2227631/rs11178), ‘5G5G/TC’ (rs1799889/rs11178), ‘5G5G/CC’ (rs1799889/rs11178), ‘5G5G/AA’ (rs1799889/rs1050955), ‘GA/AA’ (rs6092/rs1050955), ‘GG/GA’ (rs2227694/rs1050955), ‘GG/AA’ (rs2227694/rs1050955), ‘TC/GA’ (rs11178/rs1050955), and ‘CC/AA’ (rs11178/rs1050955) (Table 5).

In contrast, the following PAI-1 alleles were associated with decreased stroke risk: ‘AA/TT’ (rs2227631/rs11178), ‘4G4G/GA’ (rs1799889/rs6092), ‘GA/TT’ (rs6092/rs11178), ‘GA/GG’ (rs6092/rs7242), and ‘GA/GG’ (rs6092/rs1050955).

Before revised: Using unadjusted logistic regression analysis, an increased risk of metabolic syndrome was associated with PAI-1 alleles ‘GA/5G5G’ (rs2227631/rs1799889), ‘GA/GG’ (rs2227631/rs6092), ‘AA/GA’ (rs2227631/rs2227694) and ‘GA/AA’ (rs2227631/rs1050955), whereas decreased risk of metabolic syndrome was associated with PAI-1 alleles ‘4G4G/GA’ (rs1799889/rs6092), ‘GG/TC’ (rs6092/rs11178), ‘TC/TG’ (rs11178/rs7242), and ‘CC/TT’ (rs11178/rs7242) (Table 5).

After revised: Using unadjusted logistic regression analysis, an increased risk of metabolic syndrome was associated with PAI-1 alleles ‘GA/5G5G’ (rs2227631/rs1799889), ‘GA/GG’ (rs2227631/rs6092), ‘AA/GA’ (rs2227631/rs2227694) and ‘GA/AA’ (rs2227631/rs1050955). Conversely, decreased risk of metabolic syndrome was associated with PAI-1 alleles ‘4G4G/GA’ (rs1799889/rs6092), ‘GG/TC’ (rs6092/rs11178), ‘TC/TG’ (rs11178/rs7242), and ‘CC/TT’ (rs11178/rs7242) (Table 5).

The authors are advised to recheck the statistical values to prevent any mistakes.

We greatly appreciate your invaluable comments. While we have been responsible for the statistical analysis, we have thoroughly double-checked our work to ensure accuracy, and we thank you for your contribution.

Round 2

Reviewer 2 Report (New Reviewer)

The authors have answered all my concerns and I recommend publication

Author Response

Comments and Suggestions for Authors

The authors have answered all my concerns and I recommend publication

Thank you for taking the time to review our manuscript and for providing us with such positive feedback.

Reviewer 4 Report (New Reviewer)

The modified version of the manuscript is good and can be accepted for publication

Author Response

Comments and Suggestions for Authors

The modified version of the manuscript is good and can be accepted for publication

Thank you for taking the time to review our manuscript and for providing us with such positive feedback.

This manuscript is a resubmission of an earlier submission. The following is a list of the peer review reports and author responses from that submission.

Round 1

Reviewer 1 Report

I would like to thank the Authors of the present Manuscript and the Editors for the opportunity to provide commentary on this work.

To my understanding, the Authors performed an epidemiological association study on seven PAI-1 3’-UTR variants (considering alleles, individual genotypes and haplotypes) between a control group and a group of patients affected by several degrees of ischemic stroke. Patients also presented further pathologies. 

Following, the Authors can find my notes:

-       I would suggest that the variants studied for the PAI-1 gene could be put in a Table, with several columns reporting the variant position, rs ID, and alleles for a more accessible read.

-       All Tables indicated in the main text are missing.

-       The Authors list all the primers used in the main text (paragraph 4.2); however, it makes for an unnecessary use of space, as it would be more appropriate and visually appealing to put them in a Table.

-       The Authors could find a way to make the text much more readable and accessible, as it is presented as an incessant list of polymorphisms with associated rs IDs, alleles end genotype sequencies that make it much less appealing.

-       Were controls matched for age and sex to the cases?

-       Other physiological states / pathologies that affect the patients could have skewed the results and, consequently, their interpretation?

-       The Discussion section may be much more incisive; as interesting as the starting point is for this Manuscript, it does not really add much to the current knowledge of ischemic stroke genetics. Overall, the material presented here could be elaborated on much more. Essentially, what is presented here is a list of genotypes/haplotypes with related scores, which is not appealing to the average reader.

Author Response

Thank you.

Reviewer 2 Report

This is a case-control study aimed to investigate the association between PAI-I polymorphisms and the risk of ischemic stroke in Korean population. There are a few points need to be concerned. The major problem is that there are too many information in this study, but the authors did not systematically present the main findings of this study and why the results showed.

1.     The term of PAI-I SNPs should be unify in the results (rs number) section 2.2 (line 93-95) and table 2 (PAI-I location). It is better to notify “After multiple comparison, only subjects with PAI-I 12068 G>A AA genotype had a 1.784-fold risk of ischemic stroke when compared to patients with GG genotype”.

2.     The same problem with result section 2.3 and table 3.

3.     What is the reason for reducing one SNP at a time when the authors conduct the haplotype analysis? It’s not easy to understand the information from table 4. In addition, why does the study suddenly investigate the association between PAI-I haplotypes and risk of metabolic syndrome? The authors did not mention it in the manuscript.

4.     The number for each cells in table 5 seems strange. For example, the total number of MS patients were 235, but there were only 14 MS patients with GG-4G4G and 8 MS patients with GA-5G5G? How about the other 213 patients? Please also check the total number of study subjects in Table 6.

Author Response

Thank you.

Reviewer 3 Report

PAI-1 (gene: SERPINE1) inhibits tissue plasminogen activator (tPA) and urokinase plasmin activator (uPA) thereby preventing fibrinolysis. Elevated PAI-1 levels have been shown to be associated with venous thromboembolism and cardiovascular disease. PAI-1 levels are modulated by a number of genetic and non-genetic factors, among the genetic factors a G-deletion (rs1799889) in the SERPINE1 promoter. Compared to 5G, the 4G variant causes enhanced transcription of PAI-1. The genotype 4G/4G is associated with a significantly increased risk of venous thromboembolism and cardiovascular disease. The analysis of this frequent polymorphism and PAI-1 activity are often performed as routine tests in the context of diagnostic thrombophilia workup. Apart from the 4G variant, further variants in the PAI-1 gene were shown to be associated with the risk of venous thromboembolism. Stoke is of course a major health issue. Interestingly and despite recurrent analyses in lots of studies of different groups, the question, whether 4G/4G is associated with an enhanced risk of stroke, could not be clarified and is still discussed controversially. The situation is complicated by the fact that some authors report an even reduced risk of stroke for the 4G allele.

The authors of the manuscript aimed to reevaluate the significance of 7 PAI-1 polymorphisms and haplotypes in a collective of Korean stroke patients (574) and controls (425). Patients and controls were genotyped for 7 the target polymorphisms, haplotypes were predicted by bioinformatics and ODDs ratios and significances were calculated. ODDs ratios were adjusted for some significant cofounders.

1. Most importantly, significance calculations have not been corrected for repeated testing (7 polymorphism and several predicted haplotypes were compared in patients and controls and furthermore several subgroups). Therefore, statistics is not convincing. ODDs ratios and levels of significance are often borderline and I postulate that after corrections there will be no / very few significant results any more.

2. Several authors and meta-analyses have already examined the role of PAI-1 polymorphisms in stroke patients, but data is inconclusive to date. Just adding 5 hundred patients and controls will not dramatically change the situation, even if there might be some population differences (genetic background).

3. The study data concerning PAI-1 is restricted to genetic analyses. Several routine tests for PAI-1 activity are available. I admit that PAI-1 activity is influenced by circadian rhythm, several patient factors (e.g. age, pregnancy) and low analyte stability, but these disadvantages should be no problem in a study setting. It would be no problem to check most results for plausibility by analyzing PAI-1- activity. This would be particularly valuable to confirm the differences between the haplotypses.

4. The authors present 7 very extensive tables in the manuscript and 8 very large tbles in the supplement. All data and diverse group analyses are given in extensive detail, but authors have to workup their data to present them in a readable format. The text of the manuscript is however very short and functional interpretation is scarce. I have the feeling that the authors want to leave a large part of the analysis and interpretation of their data to the reader. I do not think this is adequate.

Minor: The use of the English language is not professional. Many abbreviations (e.g. HTN, DM, tHcy, FDR-P) are not explained in the text and in some cases there are different abbreviations (e.g. MetS and MS) for the same expression. There are too many tables. The format of the tables is often confusing.

Author Response

Thank you.

Round 2

Reviewer 1 Report

I would like to thank the Authors and Editors for the opportunity to provide commentary to the revised version of this manuscript.

I greatly appreciate the huge amount of work necessary to accomodate my and other reviewers' comments, as it required a notable restructuring of several sections. Most of the redundant and tedious-to-read-through lists have been put into Tables or in the Supplementary material, which makes the paper much easier to read and extremely more enjoyable. Extensive editing has certainly improved the manuscript's content to the point that all of my concerns have been addressed appropriately.

Author Response

Thank you very much for your helpful comments and suggestions!

Reviewer 2 Report

The authors revised the manuscript point by point adequately. I have no more questions.

Author Response

Thank you very much for your helpful comments and suggestion! 

Reviewer 3 Report

I still think that the manuscript of Choi et al. is not suitable for publication. It is currently not sufficient to investigate 7 known polymorphisms in a gene, to compare cases and controls and to calculate significance levels for the association with a phenotype, especially if there are already studies on all polymorphisms, which present contradictory results. It is not sufficient to simply examine 500 more patients and controls from one ethnicity. In addition, I question the robustness of the statistical methods. The very numerous tables and statistical calculations are very confusing. The material and methods part is also of little help. Finally, in each of the numerous evaluations (sub-collectives, etc.) an association with another polymorphism becomes significant.
What I actually consider quite interesting are the studies on the haplotypes. But again: A purely statistical analysis is not enough. Today, at least the replication of the results in a second collective is usually required. In addition, there is an easy to determine routine parameter, the PAI-1 activity, which would be suitable to check the plausibility of the results. The latter investigations would be necessary in any case to publish the results with an impact factor of 6. Last but not least, the English language is terrible. I am not a native speaker, but the language problems are just too obvious and have not been improved a bit compared to the previous version.

Author Response

"please see the attachment"

Round 3

Reviewer 3 Report

There were actually some improvements to the manuscript in terms of language and in terms of the precision of the statements. Also the clarity has gained by the change of tables and additional figures. Unfortunately, the eperimental shortcomings, such as the lack of functional data and lack of replication of molecular genetic data, remain a major problem.